# RENDER-FM: FEEDFORWARD MODEL FOR REAL-TIME PHOTOREALISTIC VOLUMETRIC RENDERING

## ABSTRACT

Current neural volumetric rendering methods like NeRF and 3D Gaussian Splatting (3DGS) achieve photorealistic quality but require prohibitive per-scan optimization (30+ minutes for 3DGS, 10+ hours for NeRF), limiting clinical applicability. We propose Render-FM, a feedforward model that directly regresses 6D Gaussian Splatting parameters from CT volumes in a single 2.8-second forward pass—a 500× speedup. Our key innovation, Anatomy-Guided Priming (AGP), leverages segmentation masks and transfer functions (color and opacity) to provide anatomically-informed initialization. Trained on 991 diverse CT scans, Render-FM employs a 3D U-Net architecture to predict per-voxel 6DGS parameters, enabling immediate real-time rendering (328+ FPS). Experiments demonstrate that Render-FM achieves superior quality compared to optimized baselines (27.30 vs 26.63 dB PSNR), with optional 89-second fine-tuning reaching 31.67 dB PSNR. Unlike per-scan methods, Render-FM generalizes to unseen anatomies, novel transfer functions, and compositional organ visualization without retraining. This advancement transforms clinical volumetric visualization, reducing preparation time from hours to seconds while maintaining or exceeding state-of-the-art quality.

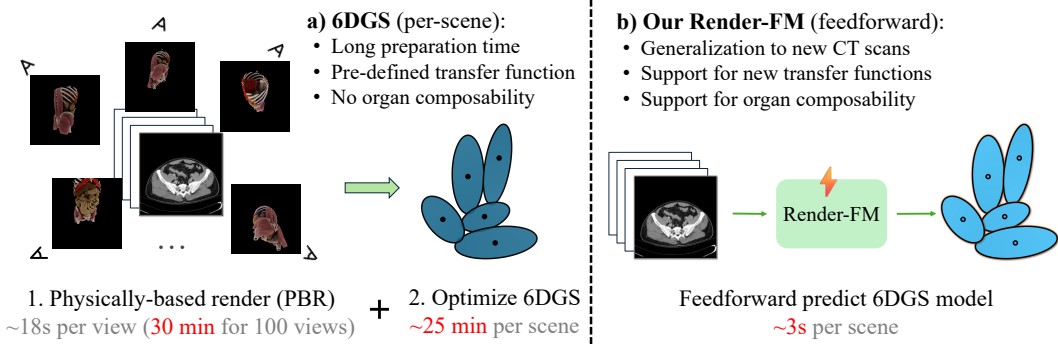

a) **6DGS** (per-scene):
• Long preparation time
• Pre-defined transfer function
• No organ composability

b) **Our Render-FM** (feedforward):
• Generalization to new CT scans
• Support for new transfer functions
• Support for organ composability

1. Physically-based render (PBR)
~18s per view (30 min for 100 views) + 2. Optimize 6DGS ~25 min per scene

Feedforward predict 6DGS model
~3s per scene

Figure 1: Pipeline comparison of a) 6DGS with per-scan optimization and b) Our Render-FM with feed-forward prediction.

## 1 INTRODUCTION

Medical imaging modalities like Computed Tomography (CT) produce rich volumetric datasets that are conventionally reviewed as sequences of 2D slices. While fundamental to clinical practice, this slice-by-slice approach often fails to convey the intricate spatial relationships of 3D anatomical structures and pathologies, particularly in complex cases involving multi-organ interactions or vascular networks (Dappa et al., 2016). Volumetric rendering addresses this limitation by synthesizing comprehensive 3D views that enable intuitive and interactive exploration, significantly enhancing diagnostic assessment, surgical planning, and patient communication (Beyer et al., 2007; Eid et al., 2017). The ability to dynamically inspect patient anatomy from arbitrary perspectives fundamentally transforms how clinicians interact with medical imaging data (Caton Jr et al., 2020).

Despite technological advancements, achieving photorealistic volumetric rendering faces significant barriers to routine clinical adoption. While conventional Direct Volume Rendering (DVR) methods

(Jung et al., 2019) enable real-time interaction through GPU-accelerated ray-casting, they employ simplified local illumination models that produce visually limited results. In contrast, photorealistic rendering techniques like Cinematic Rendering (Dappa et al., 2016; Eid et al., 2017; Wollschlaeger et al., 2020) and physically-based path tracing achieve remarkable visual quality through global illumination, soft shadows, and subsurface scattering—effects critical for realistic tissue appearance and enhanced spatial perception in surgical planning and patient communication. However, these methods require computationally expensive light transport simulation. More recent neural rendering approaches, such as Neural Radiance Fields (NeRF) (Mildenhall et al., 2021) and 3D Gaussian Splatting (3DGS) (Kerbl et al., 2023), can achieve photorealistic quality but impose prohibitive per-scan optimization requirements that fundamentally conflict with clinical time constraints.

The core limitation lies in their per-scene optimization requirement: 3DGS typically requires 30+ minutes per scan, while NeRF optimization can extend to 10+ hours. This challenge is compounded by the prerequisite generation of ground-truth views via physically-based rendering—consuming approximately 18 seconds per view or 30 minutes for 100 views. The complete preparation pipeline from raw CT data to interactive visualization thus requires nearly an hour per case, making these methods impractical for time-sensitive clinical workflows. Furthermore, optimized parameters rarely transfer between scans due to variations in anatomy, pathology, and acquisition protocols, and cannot generalize to new transfer functions (*i.e.*, color and opacity definitions) or compositional visualization capabilities essential for comprehensive diagnostic workflows.

Recent advances in large-scale feedforward models present a transformative paradigm for addressing these limitations. Inspired by large-scale pre-training principles (Bommasani et al., 2021; Moor et al., 2023; Paschali et al., 2025), Large Gaussian Models (LGMs) (Tang et al., 2024; Xu et al., 2024) have successfully applied feedforward prediction to 3D reconstruction, directly predicting Gaussian Splatting parameters from sparse image inputs and learning generalizable 3D priors from large datasets. This suggests that a similar feedforward approach could enable medical volumetric rendering by learning universal anatomical priors that generalize across patients and imaging conditions, eliminating prohibitive per-scan optimization.

We introduce Render-FM, a feedforward model for photorealistic volumetric rendering through direct prediction of 6D Gaussian Splatting (6DGS) parameters from CT volumes. Unlike optimization-based methods requiring extensive per-case training, Render-FM performs parameter regression in a single 2.8-second forward pass—achieving a 500× speedup while maintaining superior visual quality. Our approach leverages 6DGS's view-dependent modeling capabilities (Gao et al., 2024b), which capture complex optical effects at tissue interfaces crucial for photorealistic medical visualization. The model employs an encoder-decoder architecture inspired by nnU-Net's medical imaging principles (Isensee et al., 2021; 2024), combined with our novel Anatomy-Guided Priming (AGP) that incorporates anatomical segmentation and transfer function information as structured priors.

Trained on 991 diverse CT scans spanning multiple institutions and pathologies, Render-FM learns robust generalization across clinical imaging conditions. The resulting model enables immediate real-time rendering at 328+ FPS while supporting dynamic transfer function modification and compositional organ visualization—capabilities impossible with optimization-based approaches. Optional fine-tuning can further enhance quality within 89 seconds, providing flexibility while maintaining practical deployment timelines. Our key contributions include:

- **Feedforward Model Architecture:** A novel feedforward model integrating nnU-Net's medical imaging principles with 6DGS's view-dependent rendering capabilities to directly regress 6DGS parameters from CT volumes, enabling photorealistic medical volumetric visualization while eliminating per-scan optimization.
- **Anatomy-Guided Priming (AGP):** A novel initialization strategy that leverages segmentation masks and transfer functions to provide anatomically-informed structural and functional priors for 6D Gaussian primitives, enabling more effective parameter prediction.
- **End-to-End Training & Evaluation:** A comprehensive training methodology using differentiable 6DGS rendering on large-scale CT datasets with extensive evaluation demonstrating superior rendering quality, 500× speedup (hours to seconds), and robust generalizability to unseen anatomies, novel transfer functions, and compositional organ visualization.

## 2 RELATED WORK

**Volumetric Rendering in Medical Imaging** Direct Volume Rendering (DVR) synthesizes images by casting rays through volumetric data using transfer functions to map voxel intensities to optical properties (Heng & Gu, 2006; Jung et al., 2019). Standard GPU-accelerated DVR implementations enable real-time interaction in clinical viewers like 3D Slicer and OsiriX (Beyer et al., 2007), but employ simplified local illumination (e.g., Phong shading) producing visually limited results. Cinematic Rendering (Dappa et al., 2016; Eid et al., 2017; Elshafei et al., 2019) achieves photorealism through physically-based rendering with global illumination, soft shadows, and subsurface scattering. However, such photorealistic methods require expensive light transport simulation (seconds per view) and expert TF tuning, limiting routine clinical adoption despite superior visual quality.

**Neural Radiance Fields (NeRF) and 3D Gaussian Splatting** Neural Radiance Fields (NeRF) (Mildenhall et al., 2021) implicitly represent scenes using multi-layer perceptrons (MLPs) that map 5D coordinates—comprising position and view direction—to density and color. While NeRF achieves state-of-the-art view synthesis, it requires extensive per-scene optimization (often spanning hours or days), dense view inputs, and suffers from slow rendering speeds. In contrast, 3D Gaussian Splatting (3DGS) (Kerbl et al., 2023) explicitly models scenes with 3D Gaussian primitives characterized by position, covariance, opacity, and view-dependent color via spherical harmonics. Its differentiable tile-based rasterizer enables real-time rendering after optimization, though it still demands significant per-scene optimization time (typically exceeding 30 minutes). Both NeRF and 3DGS have been adapted for medical applications, including sparse-view reconstruction (Cai et al., 2024) and X-ray visualization (Gao et al., 2024a), yet their mandatory per-scene optimization hinders clinical adoption.

**6D Gaussian Splatting** 6D Gaussian Splatting (6DGS) (Gao et al., 2024b) extends 3DGS by representing primitives within a 6D spatio-angular space, characterized by a 6D covariance matrix that captures variance in both 3D position and 3D direction. This advanced formulation allows for explicit modeling of complex view-dependent effects, such as anisotropic reflections and intricate scattering phenomena frequently observed at tissue interfaces in medical data. During rendering, the 6D covariance is dynamically "sliced" based on the viewing direction to yield an effective 3D Gaussian for rasterization. 6DGS can achieve higher fidelity, potentially with fewer primitives than 3DGS (Gao et al., 2024b), making it particularly well-suited for high-quality medical visualization.

**Feedforward Models and Large Gaussian Models** Foundation models are large-scale AI systems pre-trained on vast datasets, demonstrating robust generalization to downstream tasks without requiring task-specific optimization (Bommasani et al., 2021; Moor et al., 2023; Paschali et al., 2025). Recently, Large Gaussian Models (LGMs) (Tang et al., 2024; Xu et al., 2024) have successfully applied feedforward prediction to 3D Gaussian Splatting, learning generalizable 3D scene priors from large-scale datasets. These models enable novel view synthesis from sparse 2D image inputs through direct feedforward prediction of Gaussian parameters, effectively bypassing the conventional per-scene optimization bottleneck. Render-FM adapts this feedforward paradigm to medical volumetric rendering, uniquely leveraging complete dense 3D volumetric data as input instead of sparse 2D views, and incorporating medical domain knowledge through anatomy-guided priming.

**nnU-Net Framework** The nnU-Net framework (Isensee et al., 2021; 2024) excels in medical image segmentation through automated pipeline configuration based on dataset properties. Its self-configuring 3D U-Net architecture consistently achieves state-of-the-art results across diverse benchmarks. While originally designed for segmentation, its principles of robust architecture and automated adaptation translate effectively to other medical imaging tasks (Ertl et al., 2025; Isensee et al., 2025). Render-FM leverages nnU-Net's architectural design and training robustness for our CT-to-6DGS parameter regression task.

## 3 METHODOLOGY

We present Render-FM, a feedforward model that learns a direct mapping from CT volumetric data to 6D Gaussian Splatting (6DGS) parameters, enabling real-time photorealistic rendering without per-scan optimization.

**Problem Formulation** Given a 3D CT scan $V \in \mathbb{R}^{C \times D \times H \times W}$ with $C = 6$ input channels, our goal is to predict parameters $\Theta$ defining a 6DGS representation that enables high-quality, view-

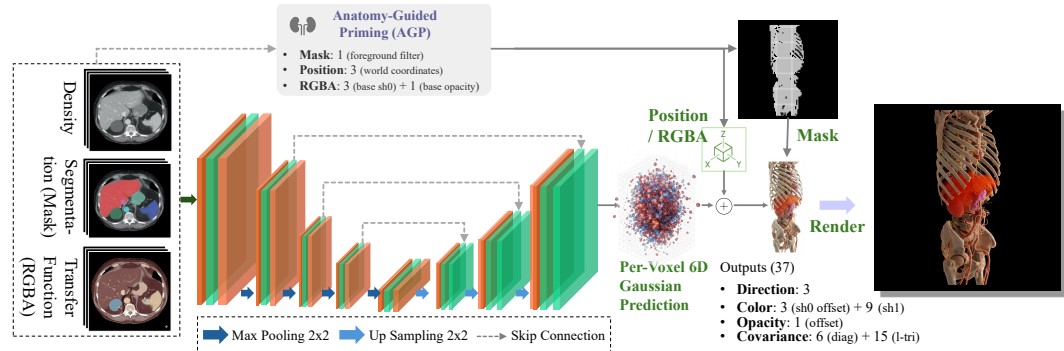

Figure 2: Overview of the Render-FM pipeline. A 3D U-Net encoder-decoder network processes a 6-channel input volume and regresses 37-channel 6D Gaussian Splatting (6DGS) parameters per voxel. Foreground voxels (mask) instantiate Gaussians. A differentiable 6DGS renderer, incorporating view-dependent slicing, produces the final image, enabling end-to-end training via rendering loss.

dependent rendering from arbitrary camera viewpoints. Formally, we learn a function $f_\phi : V \mapsto \Psi$, where $\Psi$ represents voxel-wise 6DGS parameters across the volume, parameterized by network weights $\phi$. During rendering, a subset $\Theta \subset \Psi$ corresponding to relevant foreground voxels is used to instantiate Gaussian primitives.

## 3.1 MODEL ARCHITECTURE

**Input Representation** The input volume $V$ consists of 6 channels that provide complementary information: normalized CT intensity (Hounsfield Units), segmentation mask identifying structures of interest, and 4 RGBA channels derived from pre-defined transfer functions. This multi-channel design enriches the network with information about tissue density, semantic context, and base appearance, which helps in generating more accurate Gaussian parameters.

**Anatomy-Guided Priming** Conventional initialization methods for 3DGS (Kerbl et al., 2023) rely on sparse point clouds from Structure from Motion (SfM) (Schönberger & Frahm, 2016) or random initialization, which are suboptimal for volumetric medical data as they fail to leverage domain-specific anatomical information. While recent approaches like DDGS-CT (Gao et al., 2024a) and 6DGS (Gao et al., 2024b) employ the marching-cubes algorithm to utilize CT radiodensity, they still neglect valuable segmentation, color, and opacity information. We introduce Anatomy-Guided Priming (AGP) as a comprehensive prior for 6D Gaussian primitives. Since Render-FM predicts a 6D Gaussian for each voxel, we establish position parameters directly from world coordinates of voxel indices. We derive base color and opacity values from the pre-defined transfer functions, while using segmentation masks to determine semantic classification of each Gaussian and to selectively instantiate primitives only in foreground regions. This anatomically-informed prior provides a more effective starting point that incorporates both structural and functional characteristics of the medical volume and can be seamlessly interpreted into our feed-forward pipeline.

**Encoder-Decoder Backbone** Render-FM employs a 3D U-Net architecture inspired by nnU-Net (Isensee et al., 2021). The encoder path consists of a series of blocks, each containing two $3 \times 3 \times 3$ convolutional layers with Instance Normalization, Leaky ReLU activation, and $2 \times 2 \times 2$ downsampling operations. The decoder follows a mirrored expansive path with transposed convolutions for $2 \times 2 \times 2$ upsampling and skip connections that concatenate encoder features at corresponding resolutions (excluding the final stage to output half of the input resolution). The final layer consists of a $1 \times 1 \times 1$ convolution that maps features to 37 output channels per voxel at half the input resolution ($\Psi \in \mathbb{R}^{37 \times \lfloor D/2 \rfloor \times \lfloor H/2 \rfloor \times \lfloor W/2 \rfloor}$).

**6DGS Parameter Prediction** The final layer of the decoder outputs a dense parameter volume $\Psi \in \mathbb{R}^{37 \times \lfloor D/2 \rfloor \times \lfloor H/2 \rfloor \times \lfloor W/2 \rfloor}$. Each spatial location in $\Psi$ corresponds to a potential 6D Gaussian primitive, whose 3D position $\mu_p$ is explicitly derived from the voxel's world-space coordinates. The 37 output channels at each location encode the remaining attributes defining the Gaussian's properties. Specifically, these channels represent the mean direction $\mu_d$ (3 channels), view-dependent color information encoded via Spherical Harmonic coefficients of degree $L = 0$ offset and degree $L = 1$ (12 channels total), an opacity offset (1 channel), and the components needed to construct the $6 \times 6$

covariance matrix $\Sigma$ (21 channels). To ensure a valid covariance matrix, the 6 diagonal elements are predicted in log-space, while the 15 unique off-diagonal elements are passed through a $\tanh$ activation (scaled to [-1, 1]) before reconstruction.

## 3.2 DIFFERENTIABLE 6D GAUSSIAN RENDERING

**Gaussian Instantiation** To generate the final set of Gaussians $\Theta$ for rendering, we filter potential primitives using the input segmentation mask, creating Gaussians only for voxels marked as foreground. For each instantiated Gaussian $i$, we derive its properties as follows: 1) position $\mu_{p,i}$ is set to the voxel's world coordinates; 2) color coefficients $c_i$ are computed by combining base RGB values from the input channels with predicted color offsets and spherical harmonic coefficients; 3) opacity $\alpha_i$ is calculated by applying a sigmoid function to the sum of the base alpha value and the predicted opacity offset, ensuring it remains within $[0, 1]$; 4) class label $l_i$ is assigned with the segmentation mask label; and 5) direction $\mu_{d,i}$ and covariance components are directly extracted from the corresponding location in our parameter volume $\Psi$. We then reconstruct the complete $6 \times 6$ covariance matrix $\Sigma_i$ using Cholesky decomposition to guarantee its positive semi-definiteness. This focused instantiation approach efficiently represents only the clinically relevant anatomical structures identified by the segmentation mask.

**Covariance Slicing for View-Dependence** For rendering from a specific camera viewpoint $\mathbf{p}$, we apply view-dependent covariance slicing to each instantiated Gaussian $i \in \Theta$ as described in (Gao et al., 2024b). This involves calculating the view direction $\mathbf{v}_i = (\mu_{p,i} - \mathbf{p})/||\mu_{p,i} - \mathbf{p}||$. The $6 \times 6$ covariance matrix $\Sigma_i$ is partitioned into spatial ($pp$), directional ($dd$), and cross-term ($pd$, $dp$) blocks: $\Sigma_i = \begin{pmatrix} \Sigma_{pp} & \Sigma_{pd} \\ \Sigma_{dp} & \Sigma_{dd} \end{pmatrix}$. We then compute the view-conditional 3D spatial properties: the adjusted mean $\mu'_{p,i} = \mu_{p,i} + \Sigma_{pd}\Sigma_{dd}^{-1}(\mathbf{v}_i - \mu_{d,i})$ and the effective 3D covariance $\Sigma'_{pp,i} = \Sigma_{pp} - \Sigma_{pd}\Sigma_{dd}^{-1}\Sigma_{dp}$. An opacity modulation factor $w_i = \mathcal{N}(\mathbf{v}_i|\mu_{d,i}, \Sigma_{dd})$, derived from the directional component, is also calculated. This slicing adapts each Gaussian's shape and opacity based on the viewing angle, capturing complex view-dependent effects crucial for realism.

**Tile-Based Rasterization** The resulting view-dependent 3D Gaussians (defined by $\mu'_{p,i}$, $\Sigma'_{pp,i}$, modulated opacity $w_i\alpha_i$, and view-dependent color evaluated from $c_i$ and $\mathbf{v}_i$) are rendered using a differentiable tile-based rasterizer, adapted from (Kerbl et al., 2023). Gaussians are projected onto the image plane, sorted by depth, culled, and assigned to screen tiles. Alpha compositing is performed efficiently within each tile in parallel on the GPU. This differentiable process allows end-to-end training and achieves real-time rendering speeds after the initial network inference.

## 3.3 TRAINING METHODOLOGY

**Loss Function** We train Render-FM end-to-end with the same rendering-based losses as 3DGS (Kerbl et al., 2023) that capture different aspects of visual quality:

$$\mathcal{L} = \lambda_{L1}\mathcal{L}_{L1} + \lambda_{SSIM}\mathcal{L}_{SSIM}. \tag{1}$$

The loss components include $\mathcal{L}_{L1} = ||\hat{I} - I_{gt}||_1$, which measures the absolute difference between rendered and ground-truth images, and $\mathcal{L}_{SSIM} = 1 - \text{MS-SSIM}(\hat{I}, I_{gt})$, which captures structural similarity using SSIM (Wang et al., 2004a). This combination of losses ensures that the model optimizes for both pixel-level accuracy and perceptual quality.

**Training Protocol** Our training procedure operates by first processing the input volume $V$ through the network to obtain the parameter volume $\Psi$. We then instantiate 6D Gaussians $\Theta$ only at foreground voxel locations identified by the segmentation mask. For each training iteration, we sample random camera viewpoints $\mathbf{p}$ and render the predicted Gaussians through differentiable rasterization $\hat{I} = R(\Theta, \mathbf{p})$. The rendered images are compared with ground-truth renderings $I_{gt}$ using our combined loss function, and gradients are backpropagated through both the renderer and network.

## 3.4 INFERENCE PIPELINE

During inference, Render-FM processes a new CT volume in a single forward pass, drastically reducing the time from raw data to interactive visualization. The pipeline begins with preprocessing

Table 1: Quantitative comparison. *ID*: in-domain (TotalSegmentator); *OOD*: out-of-domain (CT-ORG); *Seen/Unseen TF*: transfer functions used/not used for training; *AGP*: anatomy-guided priming; *Skeleton group*: compositional visualization requiring 0.0s additional preparation.

| Dataset | Type | Method | SSIM | PSNR | LPIPS | Time (s) | # points | FPS |
|---|---|---|---|---|---|---|---|---|
| TotalSeg | *ID* *Seen TF* | 6DGS | 0.912 | 26.63 | 0.096 | 1463.9 | 68,785 | 697.5 |
| | | 6DGS + AGP (**Ours**) | 0.925 | 28.92 | 0.093 | 1786.5 | 135,827 | 575.6 |
| | | Render-FM (**Ours**) | 0.919 | 27.30 | 0.097 | **2.8** | 343,058 | 328.6 |
| | | Render-FM (**Ours**) + FT | **0.937** | **31.67** | **0.088** | 89.4 | 227,925 | 423.8 |
| CT-ORG | *OOD* *Seen TF* | 6DGS | 0.903 | 25.97 | 0.105 | 1528.7 | 75,956 | 679.1 |
| | | 6DGS + AGP (**Ours**) | 0.926 | 29.36 | 0.091 | 2261.9 | 239,609 | 411.0 |
| | | Render-FM (**Ours**) | 0.918 | 26.21 | 0.092 | **2.6** | 586,225 | 245.2 |
| | | Render-FM (**Ours**) + FT | **0.940** | **32.48** | **0.082** | 136.2 | 469,969 | 275.1 |
| CT-ORG | *OOD* *Unseen TF* | 6DGS | 0.908 | 26.59 | 0.103 | 1546.4 | 81,802 | 684.3 |
| | | 6DGS + AGP (**Ours**) | 0.924 | 29.66 | 0.090 | 2129.3 | 246,017 | 385.5 |
| | | Render-FM (**Ours**) | 0.913 | 26.77 | 0.092 | **2.8** | 586,225 | 245.5 |
| | | Render-FM (**Ours**) + FT | **0.936** | **31.91** | **0.083** | 133.4 | 462,219 | 277.5 |
| CT-ORG | *OOD* *Seen TF* *Skeleton group* | 6DGS | 0.935 | 26.78 | 0.066 | 5146.0 | 76,848 | 602.0 |
| | | 6DGS + AGP (**Ours**) | 0.938 | 28.95 | 0.064 | 7545.3 | 286,308 | 425.1 |
| | | Render-FM (**Ours**) | 0.925 | 26.10 | 0.070 | **0.0** | 586,225 | 295.7 |
| | | Render-FM (**Ours**) + FT | **0.944** | **30.74** | **0.061** | 140.1 | 466,638 | 337.3 |

the input by normalizing intensities and generating transfer function-based RGBA values. If a segmentation mask is unavailable, we apply a pre-trained segmentation model (*e.g.*, TotalSegmentator (Wasserthal et al., 2023)). The preprocessed volume is then passed through Render-FM (taking approximately less than 3 seconds on a GPU), and the predicted parameters are filtered using the segmentation mask to instantiate 6D Gaussians. Once initialized, these Gaussians enable immediate, real-time interactive rendering. The entire process from raw CT data to interactive visualization takes mere seconds rather than approximately an hour required by optimization-based approaches while maintaining comparable or better visual quality. We can also finetune the output 6DGS model for less than 2 min to further improve rendering quality. This dramatic reduction in preparation time represents a transformative improvement for clinical workflows where timely visualization is crucial.

## 4 EXPERIMENTS

We conduct extensive experiments to evaluate the performance of Render-FM against the state-of-the-art volumetric rendering method – the 6DGS baseline (Gao et al., 2024b), focusing on rendering quality, computational efficiency, and generalizability across diverse settings. Our experiments assess Render-FM's ability to produce high-fidelity, real-time visualizations without per-scan optimization. We also verify the effectiveness of anatomy-guided priming (AGP), explore fine-tuning to enhance performance, and demonstrate Render-FM's utility in compositional organ visualization.

### 4.1 EXPERIMENTAL PROTOCOL

**Datasets** We utilized two publicly available CT datasets: 1) **TotalSegmentator (Wasserthal et al., 2023):** This dataset comprises 1,228 CT scans from clinical routines, covering 117 anatomical classes. It includes a wide range of pathologies, scanners, acquisition protocols, and institutions, making it representative of real-world clinical variability. After filtering scans exceeding 48 million voxels (due to GPU memory constraints) or lacking orthonormal directionality, we used 991 scans for training and 46 for in-domain (*ID*) testing; 2) **CT-ORG (Rister et al., 2020):** This dataset includes 140 CT scans from diverse sources, featuring both large organs (*e.g.*, lungs) and small, challenging structures (*e.g.*, bladder). We selected 10 scans (volumes 2–11) for out-of-domain (*OOD*) testing to evaluate Render-FM's generalizability to unseen data distributions.

**Data Preparation** To prepare the training data, we applied a standardized preprocessing pipeline: 1) *Resampling:* Volumes were resampled to isotropic spacing of $1.5\,\mathrm{mm}$ in all dimensions to ensure consistency; 2) *Normalization:* CT intensities (Hounsfield Units) were normalized following the nnU-Net pipeline (Isensee et al., 2021) to standardize intensity ranges; 3) *Segmentation:* Segmentation masks were generated using TotalSegmentator (Wasserthal et al., 2023), grouping 117 anatomical

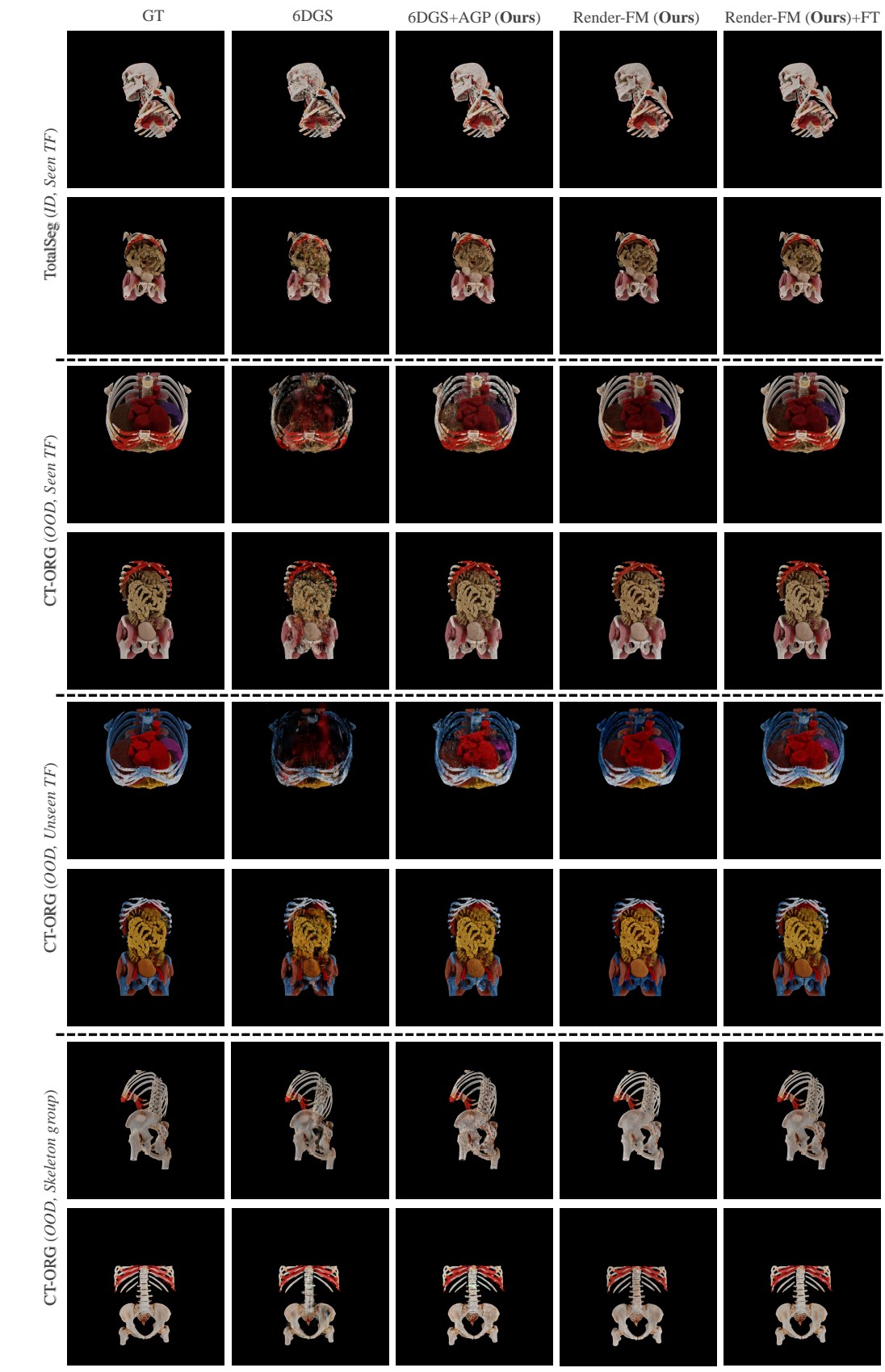

Figure 3: Qualitative comparison of rendering methods (zoom in for more details).

classes into 11 semantic categories (*e.g.*, skeleton, muscle, cardioVascular; see Supplementary Material) – note, masks can also be either manually annotated or automatically generated by other methods, ensuring flexibility for clinical use; 4) *Transfer Functions:* We defined RGBA transfer functions for the 11 semantic groups to provide base appearance cues, enhancing the model's ability to predict Gaussian parameters; and 5) *Ground-Truth Rendering:* For each training scan, we rendered 60 views of all classes, 30 views of skeleton group, and 30 views of organ groups (except the skeleton and muscle groups) with the resolution of $1600 \times 1600$ using physically-based rendering via PBRT (Pharr et al., 2023) to serve as ground truth, which takes 18.8 seconds per view on average. To enhance model robustness, we applied data augmentations, including random intensity shifts, Gaussian noise, and simulated acquisition artifacts, mimicking variations in clinical CT imaging.

For evaluation, we applied identical preprocessing to both test sets: 46 TotalSegmentator scans (*ID*) and 10 CT-ORG scans (*OOD*). Our experiments evaluated performance under three conditions: 1) *Seen TF*: using transfer functions identical to training for testing generalization to new scans; 2) *Unseen TF*: using novel transfer functions not seen during training to test appearance generalizability; and 3) *Skeleton group*: visualizing only skeletal structures to evaluate compositional capabilities. For each test scan, we rendered 40 views for computing evaluation metrics, while 20 views were used for training 6DGS baseline or fine-tuning Render-FM.

For semantic labeling of the 6DGS baseline, the original implementation does not support per-Gaussian class assignment. We therefore extended it using $k$-nearest neighbors (with $k$=1) to assign anatomical classes from the segmentation mask to each Gaussian primitive during initialization. In contrast, our anatomy-guided priming approach (6DGS + AGP) integrates semantic classification directly during the initialization process, providing more consistent anatomical structure representation.

**Implementation Details** Render-FM was implemented in PyTorch using the Adam optimizer (Kingma & Ba, 2014) ($\beta_1 = 0.9$, $\beta_2 = 0.999$) with an initial learning rate of $1 \times 10^{-3}$ and PolyLR scheduling (Chen et al., 2017). Due to GPU memory constraints, we used a batch size of 3 volumes, leveraging automatic mixed precision (AMP) and gradient accumulation for efficiency. Training was performed on a single NVIDIA A100 80GB GPU, requiring approximately 3 days. We adopt FlashGS (Feng et al., 2024) for the rasterization to further improve the real-time rendering at the inference stage. The number of instantiated Gaussians varied by scan complexity, ranging from 50,000 to 800,000, depending on the anatomical structures present.

The 6DGS experiments, including the orignal 6DGS and 6DGS with our AGP initialization (*i.e.*, 6DGS + AGP), were trained following the official implementation (Gao et al., 2024b) for 30,000 iterations, with evaluations every 500 iterations to mitigate potential overfitting when training with limited views (20 per scan). We reported the best results for 6DGS experiments to ensure fair comparison. For Render-FM fine-tuning (*FT*), we conducted 300 optimization iterations, representing a substantial reduction in computational requirements, and reported the final results.

**Evaluation Metrics** To assess rendering quality, we employed three standard metrics: Peak Signal-to-Noise Ratio (PSNR), which quantifies pixel-level accuracy with higher values indicating better fidelity; Structural Similarity Index (SSIM) (Wang et al., 2004b), which evaluates structural and perceptual similarity; and Learned Perceptual Image Patch Similarity (LPIPS) (Zhang et al., 2018), which measures perceptual similarity with lower values indicating closer alignment to human visual perception. For efficiency, we measured preparation time (in seconds), defined as the duration from raw CT input to 6DGS interactive rendering readiness; the number of Gaussian points, which reflects model complexity; and frames per second (FPS), which quantifies real-time rendering performance.

## 4.2 COMPARISON WITH BASELINE

Table 1 summarizes the performance of Render-FM, the 6DGS baseline (Gao et al., 2024b), 6DGS with anatomy-guided priming (AGP), and Render-FM with fine-tuning (*FT*) across the TotalSegmentator (*ID*) (Wasserthal et al., 2023) and CT-ORG (*OOD*) (Rister et al., 2020) datasets, under *Seen TF* and *Unseen TF* and *Skeleton group* conditions. Results highlight Render-FM's superior efficiency and rendering quality compared to the baseline.

**In-Domain (*TotalSeg, Seen TF*)** On the TotalSegmentator test set with seen transfer functions, Render-FM achieves an SSIM of 0.919, PSNR of 27.30, and LPIPS of 0.097, surpassing the 6DGS baseline (SSIM: 0.912, PSNR: 26.63, LPIPS: 0.096) without requiring per-scan optimization. Notably, Render-FM reduces preparation time from 1463.9 seconds (6DGS) to just 2.8 seconds—a 500-fold

improvement—while sustaining real-time rendering at 328.6 FPS. Fine-tuning Render-FM further elevates performance to an SSIM of 0.937, PSNR of 31.67, and LPIPS of 0.088, outperforming all methods, with a preparation time of 89.4 seconds. The 6DGS with AGP initialization improves over the baseline (SSIM: 0.925, PSNR: 28.92, LPIPS: 0.093) but requires longer optimization (1786.5 seconds) due to initializing more Gaussian points, demonstrating the benefit of leveraging our Render-FM's anatomical priors.

**Out-of-Domain (*CT-ORG, Seen TF*)**  For out-of-domain testing on the CT-ORG dataset with seen transfer functions, Render-FM exhibits robust generalization, achieving an SSIM of 0.918, PSNR of 26.21, and LPIPS of 0.092, compared to the 6DGS baseline (SSIM: 0.903, PSNR: 25.97, LPIPS: 0.105). Preparation time remains exceptionally low at 2.6 seconds, with a rendering speed of 245.2 FPS. Fine-tuning significantly enhances performance (SSIM: 0.940, PSNR: 32.48, LPIPS: 0.082), achieving the best results across all metrics. The AGP-initialized 6DGS also improves over the baseline (SSIM: 0.926, PSNR: 29.36, LPIPS: 0.091), underscoring its efficiency advantage.

**Unseen Transfer Functions (*CT-ORG, Unseen TF*)**  When evaluated with novel transfer functions on the CT-ORG dataset, Render-FM maintains strong performance (SSIM: 0.913, PSNR: 26.77, LPIPS: 0.092), demonstrating its ability to generalize to unseen appearance mappings. Fine-tuning further improves results (SSIM: 0.936, PSNR: 31.91, LPIPS: 0.083) while keeping preparation time low at 133.4 seconds, highlighting Render-FM's adaptability to diverse clinical scenarios.

**Composability (*CT-ORG, Skeleton group*)**  We further evaluated Render-FM's capability for compositional organ visualization. By selectively rendering Gaussians corresponding to specific anatomical classes (*e.g.*, lungs, liver, bones) based on the input segmentation mask, Render-FM enables interactive exploration of individual or combined structures. On the CT-ORG dataset, incorporating semantic labels in 6DGS significantly increases the training time (1528.7s vs. 5146.0s). Render-FM maintains comparable performance, demonstrating its composability without reprocessing the volume (*i.e.*, 0.0s additional preparation time). Fine-tuning also improves results (SSIM: 0.944, PSNR: 30.74, LPIPS: 0.061) with the preparation time only 140.1 seconds.

Across all conditions, Render-FM consistently delivers rendering quality comparable or better than the 6DGS baseline, with preparation times reduced by orders of magnitude. The higher number of Gaussian points in Render-FM reflects its dense parameter prediction strategy. This ensures comprehensive coverage of complex anatomies, resulting in lower FPS (245.2-423.8) compared to 6DGS (385.5–697.5), yet remains far sufficient for real-time clinical applications. Fine-tuning leverages Render-FM's robust initialization, achieving state-of-the-art quality with minimal additional computation. While AGP enhances 6DGS performance by incorporating anatomical priors, it cannot match Render-FM's feedforward efficiency, requiring extensive per-case optimization. These results underscore the transformative potential of Render-FM for clinical workflows.

Figure 3 provides a qualitative comparison, illustrating Render-FM's visual fidelity. The 6DGS baseline, trained on sparse views (40 views for composability experiments and 20 views for others), often overfits, producing floating noise artifacts in novel views. AGP-initialized 6DGS reduces these artifacts but does not eliminate them. In contrast, Render-FM leverages its learned generalizable priors to largely mitigate floating noise, though it may appear slightly blurry in some cases. After brief fine-tuning, Render-FM significantly enhances visual quality, capturing intricate details such as vasculature and bone interfaces with superior clarity compared to 6DGS, which are critical for clinical interpretation and diagnostic accuracy.

## 5 CONCLUSION

We presented Render-FM, a feedforward model for real-time, high-fidelity volumetric rendering of CT scans using 6D Gaussian Splatting. By employing an nnU-Net inspired encoder-decoder architecture trained end-to-end, Render-FM directly regresses 6DGS parameters from a multi-channel CT input volume, without the need for time-consuming per-scan optimization. The model leverages large-scale pre-training to learn generalizable mappings from CT data to expressive, view-dependent 3D representations. Our experiments demonstrate that Render-FM achieves rendering quality comparable or better than that of per-scan optimized methods while reducing preparation time from hours to seconds, enabling interactive frame rates suitable for clinical use. By integrating the robustness of nnU-Net with the expressive power and view-dependent rendering capabilities of 6DGS, Render-FM bridges the gap between advanced neural rendering quality and clinical practicality.

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

# A APPENDIX

This appendix provides additional cross-dataset validation results and implementation details for the Render-FM framework. We first present zero-shot generalization experiments on the CTPelvic1K dataset, then provide comprehensive information on anatomical label consolidation from the TotalSegmentator dataset and detailed transfer function specifications for visualization. These details are intended to enhance reproducibility and provide deeper insights into our approach.

## A.1 ADDITIONAL CROSS-DATASET VALIDATION: CTPELVIC1K

To further validate zero-shot generalization capability, we evaluate Render-FM on the CTPelvic1K dataset (Liu et al., 2021), an independent dataset from different institutions with distinct acquisition protocols and patient populations. Table 2 presents quantitative results averaged over the first 6 scans, comparing our method against 6DGS baseline and 6DGS with AGP initialization. Render-FM achieves competitive quality (SSIM: 0.926) in zero-shot inference with dramatic 2,500× speedup (0.65s vs. 1,643s for 6DGS). Note that the faster inference time (0.65s vs 2.8s for TotalSegmentator) is due to the smaller average volume size of CTPelvic1K scans compared to the TotalSegmentator and CT-ORG datasets. With brief fine-tuning (78s), it achieves best quality across all metrics (SSIM: 0.941, PSNR: 33.30) while maintaining real-time rendering (553.6 FPS). These results confirm effective cross-dataset generalization without per-scan optimization.

Table 2: Quantitative results on CTPelvic1K dataset (first 6 scans average). *AGP*: Anatomy-Guided Priming; *FT*: Fine-tuning (300 iterations). All methods use the same 6DGS renderer backend.

| Method | SSIM ↑ | PSNR ↑ | LPIPS ↓ | Time (s) ↓ | # Points | FPS ↑ |
|---|---|---|---|---|---|---|
| 6DGS | 0.893 | 27.95 | 0.155 | 1,643.32 | 45,041 | 712.4 |
| 6DGS + AGP (**Ours**) | 0.936 | 31.29 | 0.100 | 1,632.04 | 95,594 | 624.2 |
| Render-FM (**Ours**) | 0.926 | 27.15 | 0.105 | **0.65** | 164,406 | 535.7 |
| Render-FM (**Ours**) + FT | **0.941** | **33.30** | **0.096** | 78.02 | 149,243 | 553.6 |

## A.2 TOTALSEGMENTATOR LABEL MAPPING TO CONSOLIDATED GROUPS

Tables 3 and 4 detail the mapping of the original anatomical class labels from the TotalSegmentator dataset Wasserthal et al. (2023) (up to label 117) and user-defined labels (118, 119) to the 11 consolidated semantic groups used in the Render-FM framework. This grouping strategy helps in managing the complexity of anatomical structures and provides a coherent basis for applying transfer functions and training the model. The consolidated transfer function mappings are as follows:

```
// Consolidated Transfer Function Mappings:
// Index 0: Background/Other
// Index 1: Spleen
// Index 2: Liver
// Index 3: Digestive Group (Stomach, Bowels, Colon, GB, Panc, Eso)
// Index 4: Gland Group (Adrenals, Thyroid)
// Index 5: Lung Group
// Index 6: Trachea
// Index 7: Skeleton Group (Bones, Cartilage)
// Index 8: CardioVascular Group (Heart & Vessels)
```

```
// Index 9: Nervous System Group (Brain, Spinal Cord)
// Index 10: Muscle Group
// Index 11: Kidney/Urogenital Group (Kidneys, Cysts, Bladder, Prostate)
```

## A.3 TRANSFER FUNCTION DEFINITIONS

Table 5 details the RGBA transfer functions used. Each transfer function is defined by a series of points, where each point has a Hounsfield Unit (HU) value and a corresponding RGBA (Red, Green, Blue, Alpha) value. The RGBA values are typically in the range of 0-255 for colors and 0.0-1.0 for alpha (opacity).

Table 3: Mapping of TotalSegmentator and user-defined labels (Part 1: Labels 0-60) to 11 consolidated semantic groups. The 'Index' refers to the consolidated group index used in our framework.

| Original Label | Original Description (TotalSegmentator / User-defined) | Consolidated Group Index | Consolidated Group Name |
|---|---|---|---|
| 0 | Background/Other | 0 | Background/Other |
| 1 | spleen | 1 | Spleen |
| 2 | kidney_right | 11 | Kidney/Urogenital Group |
| 3 | kidney_left | 11 | Kidney/Urogenital Group |
| 4 | gallbladder | 3 | Digestive Group |
| 5 | liver | 2 | Liver |
| 6 | stomach | 3 | Digestive Group |
| 7 | pancreas | 3 | Digestive Group |
| 8 | adrenal_gland_right | 4 | Gland Group |
| 9 | adrenal_gland_left | 4 | Gland Group |
| 10 | lung_upper_lobe_left | 5 | Lung Group |
| 11 | lung_lower_lobe_left | 5 | Lung Group |
| 12 | lung_upper_lobe_right | 5 | Lung Group |
| 13 | lung_middle_lobe_right | 5 | Lung Group |
| 14 | lung_lower_lobe_right | 5 | Lung Group |
| 15 | esophagus | 3 | Digestive Group |
| 16 | trachea | 6 | Trachea |
| 17 | thyroid_gland | 4 | Gland Group |
| 18 | small_bowel | 3 | Digestive Group |
| 19 | duodenum | 3 | Digestive Group |
| 20 | colon | 3 | Digestive Group |
| 21 | urinary_bladder | 11 | Kidney/Urogenital Group |
| 22 | prostate | 11 | Kidney/Urogenital Group |
| 23 | kidney_cyst_left | 11 | Kidney/Urogenital Group |
| 24 | kidney_cyst_right | 11 | Kidney/Urogenital Group |
| 25 | sacrum | 7 | Skeleton Group |
| 26 | vertebrae_S1 | 7 | Skeleton Group |
| 27 | vertebrae_L5 | 7 | Skeleton Group |
| 28 | vertebrae_L4 | 7 | Skeleton Group |
| 29 | vertebrae_L3 | 7 | Skeleton Group |
| 30 | vertebrae_L2 | 7 | Skeleton Group |
| 31 | vertebrae_L1 | 7 | Skeleton Group |
| 32 | vertebrae_T12 | 7 | Skeleton Group |
| 33 | vertebrae_T11 | 7 | Skeleton Group |
| 34 | vertebrae_T10 | 7 | Skeleton Group |
| 35 | vertebrae_T9 | 7 | Skeleton Group |
| 36 | vertebrae_T8 | 7 | Skeleton Group |
| 37 | vertebrae_T7 | 7 | Skeleton Group |
| 38 | vertebrae_T6 | 7 | Skeleton Group |
| 39 | vertebrae_T5 | 7 | Skeleton Group |
| 40 | vertebrae_T4 | 7 | Skeleton Group |
| 41 | vertebrae_T3 | 7 | Skeleton Group |
| 42 | vertebrae_T2 | 7 | Skeleton Group |
| 43 | vertebrae_T1 | 7 | Skeleton Group |
| 44 | vertebrae_C7 | 7 | Skeleton Group |
| 45 | vertebrae_C6 | 7 | Skeleton Group |
| 46 | vertebrae_C5 | 7 | Skeleton Group |
| 47 | vertebrae_C4 | 7 | Skeleton Group |
| 48 | vertebrae_C3 | 7 | Skeleton Group |
| 49 | vertebrae_C2 | 7 | Skeleton Group |
| 50 | vertebrae_C1 | 7 | Skeleton Group |
| 51 | heart | 8 | CardioVascular Group (Heart & Vessels) |
| 52 | aorta | 8 | CardioVascular Group (Heart & Vessels) |
| 53 | pulmonary_vein | 8 | CardioVascular Group (Heart & Vessels) |
| 54 | brachiocephalic_trunk | 8 | CardioVascular Group (Heart & Vessels) |
| 55 | subclavian_artery_right | 8 | CardioVascular Group (Heart & Vessels) |
| 56 | subclavian_artery_left | 8 | CardioVascular Group (Heart & Vessels) |
| 57 | common_carotid_artery_right | 8 | CardioVascular Group (Heart & Vessels) |
| 58 | common_carotid_artery_left | 8 | CardioVascular Group (Heart & Vessels) |
| 59 | brachiocephalic_vein_left | 8 | CardioVascular Group (Heart & Vessels) |
| 60 | brachiocephalic_vein_right | 8 | CardioVascular Group (Heart & Vessels) |

Table 4: Mapping of TotalSegmentator and user-defined labels (Part 2: Labels 61-119) to 11 consolidated semantic groups. The 'Index' refers to the consolidated group index used in our framework.

| Original Label | Original Description (TotalSegmentator / User-defined) | Consolidated Group Index | Consolidated Group Name |
| --- | --- | --- | --- |
| 61 | atrial_appendage_left | 8 | CardioVascular Group (Heart & Vessels) |
| 62 | superior_vena_cava | 8 | CardioVascular Group (Heart & Vessels) |
| 63 | inferior_vena_cava | 8 | CardioVascular Group (Heart & Vessels) |
| 64 | portal_vein_and_splenic_vein | 8 | CardioVascular Group (Heart & Vessels) |
| 65 | iliac_artery_left | 8 | CardioVascular Group (Heart & Vessels) |
| 66 | iliac_artery_right | 8 | CardioVascular Group (Heart & Vessels) |
| 67 | iliac_vena_left | 8 | CardioVascular Group (Heart & Vessels) |
| 68 | iliac_vena_right | 8 | CardioVascular Group (Heart & Vessels) |
| 69 | humerus_left | 7 | Skeleton Group |
| 70 | humerus_right | 7 | Skeleton Group |
| 71 | scapula_left | 7 | Skeleton Group |
| 72 | scapula_right | 7 | Skeleton Group |
| 73 | clavicula_left | 7 | Skeleton Group |
| 74 | clavicula_right | 7 | Skeleton Group |
| 75 | femur_left | 7 | Skeleton Group |
| 76 | femur_right | 7 | Skeleton Group |
| 77 | hip_left | 7 | Skeleton Group |
| 78 | hip_right | 7 | Skeleton Group |
| 79 | spinal_cord | 9 | Nervous System Group (Brain, Spinal Cord) |
| 80 | gluteus_maximus_left | 10 | Muscle Group |
| 81 | gluteus_maximus_right | 10 | Muscle Group |
| 82 | gluteus_medius_left | 10 | Muscle Group |
| 83 | gluteus_medius_right | 10 | Muscle Group |
| 84 | gluteus_minimus_left | 10 | Muscle Group |
| 85 | gluteus_minimus_right | 10 | Muscle Group |
| 86 | autochthon_left | 10 | Muscle Group |
| 87 | autochthon_right | 10 | Muscle Group |
| 88 | iliopsoas_left | 10 | Muscle Group |
| 89 | iliopsoas_right | 10 | Muscle Group |
| 90 | brain | 9 | Nervous System Group (Brain, Spinal Cord) |
| 91 | skull | 7 | Skeleton Group |
| 92 | rib_left_1 | 7 | Skeleton Group |
| 93 | rib_left_2 | 7 | Skeleton Group |
| 94 | rib_left_3 | 7 | Skeleton Group |
| 95 | rib_left_4 | 7 | Skeleton Group |
| 96 | rib_left_5 | 7 | Skeleton Group |
| 97 | rib_left_6 | 7 | Skeleton Group |
| 98 | rib_left_7 | 7 | Skeleton Group |
| 99 | rib_left_8 | 7 | Skeleton Group |
| 100 | rib_left_9 | 7 | Skeleton Group |
| 101 | rib_left_10 | 7 | Skeleton Group |
| 102 | rib_left_11 | 7 | Skeleton Group |
| 103 | rib_left_12 | 7 | Skeleton Group |
| 104 | rib_right_1 | 7 | Skeleton Group |
| 105 | rib_right_2 | 7 | Skeleton Group |
| 106 | rib_right_3 | 7 | Skeleton Group |
| 107 | rib_right_4 | 7 | Skeleton Group |
| 108 | rib_right_5 | 7 | Skeleton Group |
| 109 | rib_right_6 | 7 | Skeleton Group |
| 110 | rib_right_7 | 7 | Skeleton Group |
| 111 | rib_right_8 | 7 | Skeleton Group |
| 112 | rib_right_9 | 7 | Skeleton Group |
| 113 | rib_right_10 | 7 | Skeleton Group |
| 114 | rib_right_11 | 7 | Skeleton Group |
| 115 | rib_right_12 | 7 | Skeleton Group |
| 116 | sternum | 7 | Skeleton Group |
| 117 | costal_cartilages | 7 | Skeleton Group |
| 118 | Coronary Arteries (User-defined) | 8 | CardioVascular Group (Heart & Vessels) |
| 119 | Pulmonary Artery (User-defined) | 8 | CardioVascular Group (Heart & Vessels) |

Table 5: Definition of Transfer Functions (Seen TF and Unseen TF). For each consolidated group, the color theme is listed, followed by points defining HU values and their corresponding RGBA color and opacity for both Seen and Unseen Transfer Functions.

| Group Index | Anatomical Group | Seen TF Point (HU) | Seen TF Value [R,G,B,A] | Unseen TF Point (HU) | Unseen TF Value [R,G,B,A] |
|---|---|---|---|---|---|
| 0 | Background/Other | *Neutral grayscale* | | *Neutral grayscale* | |
| | | -1024 | [0, 0, 0, 0] | -1024 | [0, 0, 0, 0] |
| | | 3072 | [0.0, 0.0, 0.0, 0.0] | 3072 | [0, 0, 0, 0] |
| 1 | Spleen | *Soft purple gradient* | | *Vibrant Magenta/Purple* | |
| | | -1024 | [0, 0, 0, 0] | -1024 | [0, 0, 0, 0] |
| | | -150 | [0, 0, 0, 0] | 0 | [0, 0, 0, 0] |
| | | 20 | [70, 50, 90, 0.05] | 40 | [150, 40, 130, 0.1] |
| | | 80 | [110, 80, 140, 0.2] | 100 | [190, 70, 160, 0.3] |
| | | 180 | [150, 120, 170, 0.5] | 200 | [220, 100, 190, 0.6] |
| | | 250 | [190, 160, 200, 0.7] | 300 | [240, 130, 210, 0.8] |
| | | 3072 | [220, 190, 230, 0.85] | 3072 | [255, 160, 230, 0.85] |
| 2 | Liver | *Realistic Brown Gradient* | | *Deep Red-Brown* | |
| | | -1024 | [0, 0, 0, 0] | -1024 | [0, 0, 0, 0] |
| | | -20 | [0, 0, 0, 0] | 10 | [0, 0, 0, 0] |
| | | 30 | [100, 70, 50, 0.1] | 50 | [130, 50, 30, 0.15] |
| | | 90 | [140, 100, 70, 0.3] | 120 | [160, 70, 50, 0.4] |
| | | 180 | [170, 130, 90, 0.6] | 220 | [180, 90, 70, 0.7] |
| | | 250 | [190, 150, 110, 0.75] | 300 | [195, 110, 85, 0.8] |
| | | 3072 | [210, 170, 130, 0.85] | 3072 | [210, 130, 100, 0.9] |
| 3 | Digestive Group | *Beige/Brown (Realistic)* | | *Ochre/Yellow-Orange* | |
| | | -1024 | [0, 0, 0, 0] | -1024 | [0, 0, 0, 0] |
| | | -50 | [0, 0, 0, 0] | -20 | [0, 0, 0, 0] |
| | | 20 | [170, 140, 100, 0.05] | 30 | [190, 140, 50, 0.1] |
| | | 80 | [190, 160, 120, 0.25] | 90 | [210, 160, 70, 0.3] |
| | | 180 | [210, 180, 140, 0.55] | 190 | [230, 180, 90, 0.6] |
| | | 250 | [225, 195, 155, 0.7] | 280 | [245, 200, 110, 0.75] |
| | | 3072 | [240, 210, 170, 0.85] | 3072 | [255, 220, 130, 0.8] |
| 4 | Gland Group | *Golden subtlety* | | *Muted Teal/Cyan* | |
| | | -1024 | [0, 0, 0, 0] | -1024 | [0, 0, 0, 0] |
| | | 0 | [0, 0, 0, 0] | 10 | [0, 0, 0, 0] |
| | | 30 | [160, 125, 35, 0.1] | 50 | [50, 120, 130, 0.15] |
| | | 100 | [200, 165, 70, 0.35] | 120 | [70, 150, 160, 0.4] |
| | | 200 | [220, 185, 80, 0.55] | 220 | [90, 180, 190, 0.65] |
| | | 250 | [240, 200, 90, 0.7] | 300 | [110, 200, 210, 0.75] |
| | | 3072 | [255, 225, 120, 0.75] | 3072 | [130, 220, 230, 0.8] |
| 5 | Lung Group | *Realistic Pinkish Beige* | | *Very Light Airy Blue* | |
| | | -1024 | [0, 0, 0, 0] | -1024 | [0, 0, 0, 0] |
| | | -850 | [190, 180, 180, 0.0008] | -900 | [170, 190, 210, 0.001] |
| | | -500 | [210, 200, 200, 0.0025] | -600 | [190, 210, 230, 0.003] |
| | | 0 | [230, 220, 220, 0.004] | -100 | [210, 230, 245, 0.005] |
| | | 1000 | [240, 230, 230, 0.006] | 500 | [220, 240, 255, 0.007] |
| | | 3072 | [245, 235, 235, 0.008] | 3072 | [230, 245, 255, 0.009] |
| 6 | Trachea | *Pale Beige/Pinkish* | | *Pale Lavender/Grey* | |
| | | -1024 | [0, 0, 0, 0] | -1024 | [0, 0, 0, 0] |
| | | -50 | [0, 0, 0, 0] | -80 | [0, 0, 0, 0] |
| | | 20 | [220, 200, 190, 0.1] | 0 | [180, 170, 190, 0.1] |
| | | 150 | [230, 210, 200, 0.35] | 100 | [200, 190, 210, 0.3] |
| | | 250 | [240, 220, 210, 0.5] | 200 | [220, 210, 230, 0.5] |
| | | 350 | [245, 225, 215, 0.65] | 350 | [235, 225, 245, 0.65] |
| | | 3072 | [250, 230, 220, 0.75] | 3072 | [245, 235, 255, 0.7] |
| 7 | Skeleton Group | *Ivory bone realism* | | *Cool white to steel blue gradient* | |
| | | -1024 | [0, 0, 0, 0] | -1024 | [0.0, 0.0, 0.0, 0.0] |
| | | 100.0 | [180, 30, 30, 0.1] | 100.0 | [240.0, 248.0, 255.0, 0.0] |
| | | 180 | [255.0, 215.0, 140.0, 0.6] | 180 | [176.0, 196.0, 222.0, 0.8] |
| | | 280 | [255.0, 240.0, 240.0, 0.9] | 350 | [70.0, 130.0, 180.0, 1.0] |
| | | 350 | [255.0, 255.0, 255.0, 1.0] | 3072.0 | [70.0, 130.0, 180.0, 1.0] |
| | | 3072.0 | [255.0, 255.0, 255.0, 1.0] | | |
| 8 | CardioVascular Group | *Muted Red Gradient* | | *Bright Anatomical Red* | |
| | | -1024 | [0, 0, 0, 0] | -1024 | [0, 0, 0, 0] |
| | | -50 | [0, 0, 0, 0] | 0 | [0, 0, 0, 0] |
| | | 50 | [120, 30, 30, 0.1] | 70 | [190, 20, 20, 0.2] |
| | | 150 | [160, 50, 50, 0.3] | 180 | [220, 40, 40, 0.5] |
| | | 250 | [180, 70, 70, 0.5] | 300 | [240, 60, 60, 0.75] |
| | | 400 | [200, 90, 90, 0.7] | 500 | [255, 80, 80, 0.85] |
| | | 600 | [220, 110, 110, 0.8] | 700 | [255, 120, 120, 0.9] |
| | | 3072 | [235, 150, 150, 0.85] | 3072 | [255, 150, 150, 0.95] |
| 9 | Nervous System Group | *Soft beige/yellowish hues* | | *Soft Mint Green* | |
| | | -1024 | [0, 0, 0, 0] | -1024 | [0, 0, 0, 0] |
| | | -20 | [0, 0, 0, 0] | 0 | [0, 0, 0, 0] |
| | | 10 | [175, 165, 115, 0.1] | 30 | [120, 190, 140, 0.1] |
| | | 80 | [215, 205, 155, 0.35] | 100 | [150, 220, 170, 0.35] |
| | | 200 | [230, 220, 170, 0.5] | 220 | [180, 240, 200, 0.55] |
| | | 350 | [240, 230, 180, 0.7] | 400 | [200, 250, 220, 0.7] |
| | | 600 | [245, 235, 195, 0.75] | 700 | [220, 255, 235, 0.75] |
| | | 3072 | [255, 245, 225, 0.85] | 3072 | [235, 255, 245, 0.8] |
| 10 | Muscle Group | *Realistic Muscle Pink to Beige* | | *Terracotta/Brownish-Red* | |
| | | -1024 | [0, 0, 0, 0] | -1024 | [0, 0, 0, 0] |
| | | 0 | [180, 120, 120, 0.05] | 20 | [160, 90, 70, 0.1] |
| | | 100 | [200, 140, 140, 0.25] | 120 | [180, 110, 90, 0.3] |
| | | 200 | [220, 160, 160, 0.4] | 220 | [200, 130, 110, 0.5] |
| | | 250 | [230, 170, 170, 0.55] | 350 | [215, 150, 130, 0.7] |
| | | 500 | [240, 180, 180, 0.7] | 600 | [230, 170, 150, 0.8] |
| | | 3072 | [245, 190, 190, 0.85] | 3072 | [240, 190, 170, 0.85] |
| 11 | Kidney/Urogenital Group | *Beige Pink to Tan* | | *Warm Orange/Tan* | |
| | | -1024 | [0, 0, 0, 0] | -1024 | [0, 0, 0, 0] |
| | | 0 | [200, 170, 150, 0.05] | 15 | [190, 120, 60, 0.1] |
| | | 150 | [210, 180, 160, 0.35] | 100 | [210, 145, 80, 0.35] |
| | | 250 | [220, 190, 170, 0.55] | 200 | [225, 165, 100, 0.6] |
| | | 400 | [230, 200, 180, 0.7] | 350 | [240, 185, 120, 0.75] |
| | | 600 | [235, 205, 185, 0.75] | 600 | [250, 200, 140, 0.8] |
| | | 3072 | [240, 210, 190, 0.85] | 3072 | [255, 215, 160, 0.85] |

