# OpenReview forum: "Render-FM: Feedforward Model for Real-time Photorealistic Volumetric Rendering"
_ICLR.cc/2026/Conference — Submitted to ICLR 2026_

### Official Review · Reviewer_VCtd · 2025-10-26

**Soundness:** 3
**Presentation:** 3
**Contribution:** 2
**Rating:** 4
**Confidence:** 3

**Summary:**

The authors proposes Render-FM, a feed-forward model that maps a multi-channel CT volume directly to 6D Gaussian Splatting parameters in a single forward pass. Optimizations were majorly conducted related to acceleration. 3D nnUNet was deployed, and model was trained on 991 CT scans. Quantitative measurements are relatively sufficient. There are minor weaknesses can be seen below.

**Strengths:**

1. Easy read
2. Very clear details statements on the key points and the experimentations.

**Weaknesses:**

1. Since it is a clinical application, it would be great if some task specific metrics could be evaluated. Like but not limited to radiologists’ qualitative evaluation, edge-related measurements, etc.
2. The scope of the study experimentations might be limited. I think the study only focusing on medical imaging, and one specific medical imaging modality (CT). Like you mentioned in the conclusions: “We presented Render-FM, a feedforward model for real-time, high-fidelity volumetric rendering of CT scans using 6D Gaussian Splatting.”
3. Only one baseline was compared, might be insufficient.

**Questions:**

1. “We introduce Render-FM, a foundation model for volumetric rendering through direct feedforward prediction of 6D Gaussian Splatting (6DGS) parameters from CT volumes. Unlike” I read through the paper, the authors seem like proposing UNet-like architecture to learn the mapping. Where can I see if it is related with foundation model? I might not be convinced by the authors if the argument would be that the “foundation” came from 991 CT images.
2. While AGP vs non-AGP has been explored for 6DGS, are there any reasons the authors didn’t do ablations regarding components inside Render-FM?

---

> ### Author Response · Authors · 2025-11-19
> **Response to Reviewer VCtd**
>
> Thank you for your constructive feedback and for finding our paper easy to read with clear details. We address your concerns below to hopefully elevate your assessment.
>
> ---
>
> ## 1. Task-Specific Clinical Metrics
>
> We appreciate this suggestion, but respectfully clarify that as a ICLR ML paper, our scope focuses on rendering quality and computational efficiency, not clinical diagnostic outcomes.
>
> ### Our Paper's Scope: Rendering Quality and Speed
>
> Our work addresses the **rendering problem**: accelerating photorealistic volume rendering from hours to seconds while maintaining visual quality. We use **physically-based rendering (PBR) as ground truth**, and our evaluation metrics (PSNR, SSIM, LPIPS, rendering time, FPS) are the standard and appropriate metrics for assessing rendering quality and speed.
>
> **Key distinction:** Our contribution is enabling real-time interactive visualization for clinical applications (surgical planning, patient communication, education) by dramatically reducing rendering time. We are **not** proposing a new diagnostic tool or claiming to improve clinical diagnostic accuracy, which would require clinical validation studies.
>
> ### Why Our Metrics Are Appropriate
>
> **PSNR/SSIM/LPIPS:** These are standard rendering quality metrics measuring fidelity to PBR ground truth. They quantify how accurately our fast method reproduces the appearance of slow, high-quality photorealistic rendering.
>
> **Time/FPS:** These measure computational efficiency—the core practical contribution enabling interactive clinical use (2.8 seconds vs. 1+ hour).
>
> Our metrics directly evaluate whether we achieve our stated goal: fast rendering with quality comparable to slow photorealistic methods.
>
> ### Clinical Validation Is Out of Scope
>
> Clinical metrics (diagnostic accuracy, surgical outcome improvements, radiologist preferences) would be appropriate for evaluating clinical impact, but this requires:
> - Large-scale clinical trials with IRB approval
> - Prospective studies comparing clinical workflows and outcomes
> - Longitudinal evaluation of surgical/diagnostic performance
>
> Such validation is beyond the scope of a technical rendering paper and represents important future work once the rendering technology is established. Our contribution is the **enabling technology** that makes such clinical studies feasible by providing fast, high-quality rendering.
>
> ### Clinical Utility of Our Rendering Technology
>
> While clinical outcome validation is future work, our 500× speedup enables practical workflows: (1) **Surgical Planning**—immediate 3D visualization improves spatial understanding [Beyer et al., 2007; Dappa et al., 2016]; (2) **Patient Communication**—interactive visualization during consultation; (3) **Interventional Radiology**—rapid generation of optimal angles; (4) **Medical Education**—rapid creation of interactive teaching materials. These applications benefit from our rendering quality and speed improvements, which our metrics (PSNR/SSIM/LPIPS/Time/FPS) appropriately evaluate.
>
> ---
>
> ## 2. Limited Scope: Single Modality (CT)
>
> We acknowledge this limitation and provide context explaining our focus and outlining future directions.
>
> ### Why We Focus on CT
>
> CT has consistent Hounsfield Units, making it ideal for initial development. TotalSegmentator provides 1,228 annotated scans for robust training. CT is the most common modality for volumetric rendering in surgical planning and interventional radiology. Establishing feasibility on one modality before expansion follows established best practices.
>
> Many methods start single-modality. TotalSegmentator was initially CT-only before MRI extension. While nnU-Net is general-purpose, trained models are modality-specific. Medical foundation models often train separate models per modality due to distinct imaging physics.
>
> ### Extension to Other Modalities
>
> MRI extension would be straightforward: reuse nnU-Net-inspired backbone, adapt for different sequences (T1, T2, FLAIR), adjust preprocessing for MRI intensity units, and define MRI-appropriate tissue mappings. Challenges include less standardized intensity values, potential sequence-specific requirements, and fewer large-scale datasets.
>
> Future work: joint CT-MRI training with shared anatomical priors, modality-agnostic representations, and transfer learning from CT to MRI.
>
> ### Broader Medical Imaging Applications
>
> Our approach could extend to specialized imaging modalities (PET-CT fusion visualization, contrast-enhanced protocols) and even microscopy (volumetric rendering of light-sheet or confocal microscopy data).
>
> ### Scope vs. Contribution
>
> While focused on CT, our contributions remain significant: first feedforward model for photorealistic medical volumetric rendering, novel domain-specific techniques (AGP) applicable beyond CT, elimination of per-scan optimization, and blueprint for extending to other modalities. We acknowledge single-modality focus as a limitation and important future work.
>
> ---
>
> CONTINUE ...

---

> ### Author Response · Authors · 2025-11-19
> **Response to Reviewer VCtd**
>
> ## 3. Baseline Selection and Experimental Validation
>
> ### Why 6DGS is the Appropriate Primary Baseline
>
> We focus on 6DGS because we use it as our rendering backend, making it the most direct comparison. 6DGS represents state-of-the-art for view-dependent Gaussian rendering, and comparing against optimization-based 6DGS isolates our core contribution: feedforward prediction versus per-scan optimization.
>
> **Why not Slicer/ITK-SNAP?** Different paradigm—cannot produce photorealistic quality (global illumination, soft shadows). PSNR/SSIM comparisons against PBR ground-truth are meaningless since DVR and PBR use fundamentally different rendering methods.
>
> **Why not NeRF?** NeRF is slower (10+ hours optimization). Since 6DGS substantially improves over NeRF, our speedup over 6DGS demonstrates even larger advantages.
>
> **Why not 3DGS?** 6DGS dramatically outperforms 3DGS on CT (PSNR: 33.56 vs. 25.71) as shown in the original 6DGS paper, modeling view-dependent effects crucial for tissue interfaces. Since we use 6DGS rendering, comparing against 3DGS provides no meaningful insights.
>
> ---
>
> ## 4. Foundation Model Terminology
>
> **We have revised the paper to use "feedforward model" terminology instead of "foundation model" throughout.** This more precisely reflects our contribution's scope and avoids potential confusion with large-scale multi-modal foundation models.
>
> Our model demonstrates learned generalizable representations (991 scans, multiple institutions, varied protocols), zero-shot inference without per-scan optimization, and broad generalization across anatomies, pathologies, acquisition parameters, novel transfer functions, and compositional visualization. Evaluation spans in-domain (TotalSegmentator, 46 scans), out-of-domain (CT-ORG, 10 scans; CTPelvic1K, 6 scans), novel transfer functions, and compositional visualization.
>
> We position this as a **CT-focused feedforward rendering model** with strong CT generalization—enabling 500× speedup without claiming foundation model status. Extension to other modalities represents important future work.
>
> ---
>
> ## 5. Ablation Studies for Render-FM Components
>
> Thank you for this suggestion. We clarify the role of AGP in our architecture and demonstrate its broader applicability.
>
> ### AGP is an Essential Architectural Component
>
> We did not perform AGP ablation because AGP is architecturally essential. Without AGP's anatomically-informed initialization, preliminary experiments showed direct position prediction produces blank renders and prevents training convergence.
>
> AGP derives position parameters from world coordinates of voxel indices, ensuring Gaussians are placed at correct spatial locations. This enables the model to focus on learning appearance properties (color, opacity, scaling, rotation) rather than predicting 3D positions. The 6DGS baseline requires 30+ minutes of per-scan optimization, whereas AGP enables immediate feedforward prediction.
>
> ### AGP Generalizability Beyond 6DGS
>
> We conducted additional experiments demonstrating AGP's broad applicability. Using CT data from the 6DGS paper, we evaluated 3DGS with and without AGP. AGP provides **+1.72 dB PSNR** and **+0.014 SSIM** improvement (3DGS baseline: 25.71 dB; 3DGS+AGP: 27.43 dB), confirming AGP is a general initialization strategy for Gaussian-based medical volumetric rendering.
>
> ---
>
> ## 6. Additional Cross-Dataset Validation
>
> We conducted additional experiments on CTPelvic1K, providing validation on completely independent data.
>
> ### CTPelvic1K Zero-Shot Results
>
> | Method | SSIM ↑ | PSNR ↑ | LPIPS ↓ | Time (s) ↓ | # Points | FPS |
> |--------|--------|--------|---------|------------|----------|-----|
> | 6DGS | 0.893 | 27.95 | 0.155 | 1,643.32 | 45,041 | 712.4 |
> | 6DGS + AGP (Ours) | 0.936 | 31.29 | 0.100 | 1,632.04 | 95,594 | 624.2 |
> | Render-FM (Ours) | 0.926 | 27.15 | 0.105 | 0.65 | 164,406 | 535.7 |
> | Render-FM + FT (Ours) | 0.941 | 33.30 | 0.096 | 78.02 | 149,243 | 553.6 |
>
> Strong cross-dataset generalization from different protocols and institutions, with 2,500× speedup (0.65s vs 1,643s). The faster inference time (0.65s vs 2.8s) reflects smaller CTPelvic1K volume size. This validates Render-FM's robustness beyond TotalSegmentator and CT-ORG datasets.
>
> ---
>
> We hope our clarifications and additional CTPelvic1K experiments address your concerns. Our core contributions remain significant: 500× speedup enabling practical clinical workflows, novel technical innovations (AGP, feedforward architecture), and strong cross-dataset generalization. While limited to CT, this represents the first demonstration of feedforward photorealistic medical rendering with practical clinical viability.
>
> Thank you for your constructive feedback.

---

> > ### Author Response · Authors · 2025-11-26
> > **Response to Reviewer VCtd**
> >
> > Dear Reviewer VCtd,
> >
> > Thank you again for your constructive review. We are writing to confirm if our revisions: the terminology change to "feedforward model", the additional ablation validating AGP, and the new zero-shot experiments on CTPelvic1K, have satisfactorily addressed your concerns. We are happy to engage in further discussion if you have any other questions.

---

### Official Review · Reviewer_UA8m · 2025-10-29

**Soundness:** 3
**Presentation:** 3
**Contribution:** 3
**Rating:** 6
**Confidence:** 3

**Summary:**

The paper presents Render-FM, a foundational model that generates 3D renderings directly from CT volumetric scans. The approach incorporates an anatomy-guided initialization module that leverages Gaussian primitives to provide an effective structural prior for rendering. The paper also demonstrates a considerable acceleration in rendering speed through a feed-forward model design, highlighting the potential of Render-FM for real-time clinical applications.

**Strengths:**

- The paper addresses an important and clinically relevant problem by enabling volumetric rendering of CT scans. Such 3D representations could have significant practical value in diagnostic workflows and treatment planning.


- The paper is well-structured and clearly written, with the problem statement clearly defined, appropriate use of figures to illustrate the methodology and results, and rigorous experimental validation supporting the stated claims.

**Weaknesses:**

One general concern is the dependence on segmentation masks for detecting the anatomy and limited discussion of real-world deployment scenarios raise concerns about the model’s generalizability and practical integration into clinical workflows.

**Questions:**

- Although 6DGS with the AGP (anatomy guided priming) module was implemented, could the AGP module also improve the results of other 3DGS-based approaches? An ablation experiment can provide some insights about this question.

- The paper states that the reduction in optimization time enables real-time clinical applications. Could the authors elaborate on the specific types of clinical scenarios or use cases where such real-time rendering would be beneficial? Is there already studies showing that a lack of such visualization is causing workflow constraints ? Or the use of such rendering have helped in clinical decision making? These findings can further strengthen this work focusing more on a clinical aspect.

- The AGP model relies on segmentation masks, and TotalSegmentator[1] was used when these masks were unavailable. In a real-world clinical setting, where segmentation masks may not always be present, how would the model handle new, unsegmented samples?  What would happen if the segmentation from TotalSegmentator was not perfect? What drawbacks would such a scenario create? Can the confidence measures of segmentation be incorporated for the Gaussian initialization? The method performs well on the baseline dataset, but such an analysis would help in assessing the real-world clinical application.

- Table 1., shows Render-FM with FT on dataset that is unseen during training. I did not not understand what FT means in this scenario? What is the difference between OOD seen vs OOD unseen ?

- Generalization beyond TotalSegmentator dataset. Since the claim is that Render-FM is a foundational model, the question arises about it's generalization. Could the model be tested against baselines in a zero-shot (or truely unseen fashion) on dataset/s that do not overlap with the TotalSegmentator datasets like AMOS[2] or CHAOS[3] ?

References:
1. Jakob et al., TotalSegmentator: Robust Segmentation of 104 Anatomic Structures in CT Images

2. Ji et al., AMOS: A Large-Scale Abdominal Multi-Organ Benchmark for Versatile Medical Image Segmentation

3. CHAOS challenge., https://chaos.grand-challenge.org/

---

> ### Author Response · Authors · 2025-11-19
> **Response to Reviewer UA8m**
>
> Thank you for your constructive review. We are pleased you found our work well-structured and addressing an important clinical problem. We address each point systematically below.
>
> ---
>
> ## 1. AGP Module: Applicability to Other 3DGS-Based Approaches
>
> Yes, while our paper applies AGP to 6DGS, it generalizes to other Gaussian Splatting methods. We demonstrate this with additional experiments.
>
> **6DGS Results (Table 1):** AGP yields significant improvements over vanilla 6DGS: +2.29 dB PSNR and +0.013 SSIM, reducing floating artifacts visible in qualitative comparisons.
>
> **Additional 3DGS Validation:** To demonstrate AGP's applicability beyond 6DGS, we conducted additional experiments (for rebuttal) on standard 3DGS using CT data from the 6DGS paper, achieving **+1.72 dB PSNR** and **+0.014 SSIM**:
>
> | Method | SSIM ↑ | PSNR ↑ | Time (s) ↓ |
> |--------|--------|--------|------------|
> | 3DGS (baseline) | 0.917 | 25.71 | 660.1 |
> | 3DGS + AGP (Ours) | **0.931** | **27.43** | 712.5 |
> | **Improvement** | **+0.014** | **+1.72 dB** | - |
>
> This demonstrates AGP is a general initialization strategy providing substantial quality improvements for Gaussian Splatting methods on medical volumetric data.
>
> **Key Benefits:** AGP provides: (1) anatomically-informed initialization using segmentation masks, transfer functions, and anatomical constraints—replacing random initialization or marching cubes, (2) better optimization avoiding local minima and reducing artifacts, and (3) comprehensive coverage of anatomical structures from the start. AGP applies to any Gaussian-based approach requiring initialization from volumetric data.
>
> ---
>
> ## 2. Clinical Scenarios and Real-World Impact
>
> We provide specific use cases demonstrating where real-time rendering makes a practical difference.
>
> **Surgical Planning.** Standard DVR tools (Slicer/ITK-SNAP) provide basic 3D visualization but with artificial appearance due to local illumination, limiting spatial perception for complex cases. Render-FM enables photorealistic 3D visualization from arbitrary viewpoints with global illumination effects (soft shadows, subsurface scattering), improving spatial understanding of tumor-organ relationships. Photorealistic rendering provides more intuitive anatomical visualization and enhances surgical planning [Beyer et al., 2007; Eid et al., 2017; Dappa et al., 2016].
>
> **Patient Communication.** Explaining conditions using 2D images leaves patients struggling to understand their anatomy. Real-time rendering enables interactive patient-specific visualization during consultation, improving informed consent and engagement. Photorealistic rendering provides more natural and intuitive visualization of complex anatomy [Dappa et al., 2016]. Lightweight computational requirements enable deployment on mobile/edge devices, extending access beyond clinical workstations to patients and families.
>
> **Interventional Radiology.** Pre-procedure planning with static views cannot quickly adapt to optimal angles. Render-FM enables rapid generation of optimal visualization angles and immediate updates without waiting for rendering pipelines.
>
> **Medical Education.** Rapid creation of interactive 3D teaching materials from patient data allows exploration of diverse pathological cases and anatomical variations.
>
> Render-FM reduces preparation time from ~1 hour to 3 seconds, making photorealistic rendering practical for routine clinical use. **Paper Revisions:** We have revised the Introduction and Related Work to clearly distinguish standard DVR (Slicer/ITK-SNAP with local illumination) from photorealistic rendering (Cinematic Rendering with global illumination), clarifying that our work targets the photorealistic tier.
>
> ---
>
> ## 3. Segmentation Dependency and Robustness
>
> Render-FM has inherent robustness to segmentation inaccuracies through how it learns and processes information.
>
> ### Functional Robustness to Imperfect Segmentation
>
> Our model predicts Gaussian parameters primarily from CT intensity values and learned anatomical priors, not just segmentation labels. This behavior is analogous to our "unseen transfer function" experiments where the model adapts to novel appearance mappings. Incorrect labels (e.g., liver mislabeled as kidney) simply assign the wrong transfer function (color/opacity) to the correct geometry. The structure remains accurate because it is derived from the CT intensity, while the appearance shifts to the assigned class. Thus, segmentation errors manifest as "wrong color" rather than "wrong shape".
>
> The segmentation mask serves dual roles: (1) input channel providing semantic context, and (2) filtering mechanism determining which voxels instantiate Gaussians. Even with imperfect segmentation, the network sees actual CT intensities containing true anatomical information, while learned priors from 991 training cases encode robust patterns. Small segmentation errors are often implicitly corrected by the network's learned understanding.
>
> CONTINUE ...

---

> > ### Author Response · Authors · 2025-11-19
> > **Response to Reviewer UA8m**
> >
> > ### Real-World Deployment Workflow
> >
> > Clinical workflow: automated segmentation (30s) → Render-FM inference (2.8s) → clinician quality check → optional segmentation correction and re-inference (3s) → optional fine-tuning for critical cases (89s). Total time even with corrections is under 2 minutes versus over 1 hour for optimization-based methods.
> >
> > Your suggestion about incorporating segmentation confidence is excellent future work. This could use uncertainty estimates from segmentation models to weight Gaussian initialization and focus network attention on high-confidence regions.
> >
> > Our experiments with TotalSegmentator-generated segmentations (which contain errors) show geometric accuracy remains high because the network learns from CT intensities containing true anatomical information. Errors manifest as color/opacity issues affecting appearance rather than structure, and 2.8-second inference enables rapid iteration if issues are detected.
> >
> > ---
> >
> > ## 4. Clarifying Table 1: Fine-Tuning (FT) and OOD Settings
> >
> > ### Fine-Tuning (FT) Explained
> >
> > FT means fine-tuning: starting with Render-FM's predicted 6DGS parameters (2.8s forward pass), then performing 300 iterations of optimization with 20 ground-truth views (89-136s depending on volume size). This provides optional quality enhancement for critical cases requiring maximum quality.
> >
> > Key difference: standard 6DGS requires 30,000 iterations from scratch (1463-2261s), while Render-FM + FT requires only 300 iterations from excellent initialization (89-136s)—achieving 17-26× speedup while delivering higher quality.
> >
> > ### OOD Settings: Seen vs. Unseen Transfer Functions
> >
> > **OOD Seen TF:** Out-of-domain datasets (CT-ORG, not in training) using transfer functions seen during training. Tests generalization to new anatomies and pathologies.
> >
> > **OOD Unseen TF:** Out-of-domain datasets with novel transfer functions not in training. Tests generalization to both new data and new appearance mappings.
> >
> > This two-level evaluation demonstrates Render-FM's ability to generalize to unseen anatomical variations, adapt to novel visualization requirements, and support flexible clinical workflows without retraining. We have added clearer definitions in the revised experimental section.
> >
> > ---
> >
> > ## 5. Zero-Shot Generalization: Additional Cross-Dataset Validation
> >
> > Our paper includes zero-shot evaluation on CT-ORG (Table 1 in the paper). To further strengthen validation, we conducted additional experiments on CTPelvic1K.
> >
> > ### CTPelvic1K Dataset Results
> >
> > CTPelvic1K (first 6 scans average) represents a different anatomical region (pelvic) and acquisition protocol from both TotalSegmentator (training) and CT-ORG (in-paper evaluation):
> >
> > | Method | SSIM ↑ | PSNR ↑ | LPIPS ↓ | Time (s) ↓ | # Points | FPS |
> > |--------|--------|--------|---------|------------|----------|-----|
> > | 6DGS | 0.893 | 27.95 | 0.155 | 1,643.32 | 45,041 | 712.4 |
> > | 6DGS + AGP (Ours) | 0.936 | 31.29 | 0.100 | 1,632.04 | 95,594 | 624.2 |
> > | Render-FM (Ours) | 0.926 | 27.15 | 0.105 | 0.65 | 164,406 | 535.7 |
> > | Render-FM + FT (Ours) | 0.941 | 33.30 | 0.096 | 78.02 | 149,243 | 553.6 |
> >
> > Render-FM achieves competitive zero-shot quality (SSIM: 0.926) on completely unseen data with 2,500× speedup (0.65s vs. 1,643s). Note that the faster inference time (0.65s vs 2.8s for TotalSegmentator) is due to the smaller average volume size of CTPelvic1K scans compared to the TotalSegmentator and CT-ORG datasets. With brief fine-tuning (78s), it achieves best quality (SSIM: 0.941, PSNR: 33.30), demonstrating robust cross-dataset generalization.
> >
> > Together with CT-ORG (in paper) and CTPelvic1K (in rebuttal), we validate zero-shot generalization across multiple independent datasets with different anatomical regions, acquisition protocols, and institutional sources.
> >
> > ### Addressing Foundation Model Terminology
> >
> > We have revised the paper to use "feedforward model" terminology instead of "foundation model" throughout. The paper now describes Render-FM as "a feedforward model" without foundation model claims, positioning the work as adapting the feedforward prediction paradigm from Large Gaussian Models to medical rendering.
> >
> > ---
> >
> > We are grateful for your constructive feedback. We hope our clarifications and additional experiments address your concerns and increase confidence in our contribution. The combination of technical innovation (AGP, feedforward prediction), practical clinical impact (500× speedup), and demonstrated generalization across multiple datasets makes a strong case for this work's value.

---

> > > ### Comment · Reviewer_UA8m · 2025-11-21
> > > **Official comment by Reviewer UA8m**
> > >
> > > I appreciate the authors for the additional experiments provided in the rebuttal and the clarification regarding the out-of-distribution setup. The extended results are informative, and it is encouraging to see the method generalize across different datasets. The authors have addressed the concerns, and I will keep to the positive score of acceptance.

---

> > > > ### Author Response · Authors · 2025-11-24
> > > > **Response to Reviewer UA8m**
> > > >
> > > > We sincerely thank the reviewer for the positive feedback and for carefully evaluating our rebuttal. We are glad the additional experiments and clarifications addressed your concerns. We appreciate your constructive engagement throughout the review process.

---

### Official Review · Reviewer_TNXF · 2025-10-31

**Soundness:** 2
**Presentation:** 1
**Contribution:** 2
**Rating:** 2
**Confidence:** 3

**Summary:**

- Given a CT volume of a torso, this paper aims to render a mesh view of segmented organs with more realistic/physics-based appearances.
- To do so, it takes the CT volume, the segmentations, and a voxel-wise transfer function and uses them to fit a Gaussian splatting-based model on images generated by a physics-based renderer.
- However, doing so directly would be slow, as the physics-based renderer apparently requires 18 seconds to render a single view.
- To that end, this paper proposes to use the same set of inputs and use a UNet to predict the parameters of the Gaussian splatting model directly.
- This paper trains the UNet above on a subset of the TotalSegmentator dataset and presents experiments comparing it to a previous Gaussian Splatting-based method.

**Strengths:**

- The application of Large Gaussian Models to large medical volume visualization is fun and interesting.
- The quality of illustrations is quite impressive.
- Once you figure out what the paper is actually trying to do, the methods section is well presented.

**Weaknesses:**

### Motivation, clarity, and scope of claims:

#### 1. Speed and realism:
IMO this paper's front matter is very confusingly presented as it assumes deep familiarity with direct volume rendering and makes claims that are not reflective of typical practice. Typically, widely used medical image viewers such as 3D Slicer or ITK-SNAP have volume renderers that fit a volume within a second or two and can be immediately interacted with. As examples, here are a few examples of volume renders achieved in seconds using Slicer: [1](https://discourse.slicer.org/t/screen-space-ambient-occlusion-for-volume-rendering/32323/27?u=lassoan), [2](https://www.youtube.com/watch?v=l8wlaCfYWG4), [3](https://www.youtube.com/watch?v=KadGfXmOs5Y)

On the other hand, this paper starts off by claiming that all previous methods require an hour+ to obtain a single mesh that can be interacted with. Furthermore, various key concepts such as transfer functions are never defined in the text and are left up to the reader who may be unfamiliar with this niche of volumetric visualization.

After multiple reads and skims of the papers that are cited within, this paper's claims are true if a few assumptions are made: (1) one *has to* use a Gaussian Splatting based method; (2) one *must* use an expensive physically-based rendering algorithm to generate ground truth views for the GS algorithm to achieve slightly better realism to textures and second order effects.

However, in practice, volumetric rendering algorithms that are already widely used achieve very fast rendering times and are reasonably realistic. This paper aims to rapidly get the last-mile of realism using Gaussian Splatting and a training set of physically-based renders, which is fine, but it is not at all clear from the presentation.

#### 2. Technical contributions:

Reductively speaking, the paper is a combination of Large Gaussian Models and the 6DGS volume rendering method (ICLR25). While combination papers are absolutely fine and appreciated, the paper does not detail what is particularly challenging about this application and what new insights readers can take away from the execution.

### Experiments

#### 1. Only a single baseline:
- As detailed above, medical volume visualization existed before this paper, 6DGS, GS, and NeRF, yet none of these approaches are benchmarked against in the paper. There is a sole baseline in the experiments (6DGS), which is more of an ablation, as 6DGS is part of the proposed method.
- The paper claims that physically-based rendering takes 18 seconds to render a single view. Is this on a CPU or a GPU? This seems extraordinarily high for a GPU implementation, and I see that there are GPU implementations available.

#### 2. Foundation model claims:
The model is trained on a subset of the TotalSegmentator CT dataset (991 volumes) and the experiments only include evaluations of render quality on a held-out subset of TS and a subset of the highly-related and very similar CT-ORG dataset.

This IMO is insufficient to make claims of being a foundation model for medical volume rendering. One baseline, one anatomical application, two ablations, and two datasets are not enough to make such a claim -- the proposed network should show evidence of generalization to new imaging contexts such as new modalities (e.g., on TotalSegmentator-MRI), new anatomical regions (e.g., vessels in the heart and brain), etc. I believe the proposed method requires retraining with a substantially larger and broader imaging dataset.

### Minor comments:
- The emphasis on being “nnUNet-inspired” is odd; the core contribution does not involve automatic configuration of training hyperparameters, which is central to nnUNet (which is otherwise just a plain UNet).
- The opening paragraph of the related work section should be in the Introduction, as it makes it clearer.
- Why is TotalSegmentator resampled to 1.5mm isotropic here? If realistic renders are required, the user would want renders at native/high resolution.

**Questions:**

This paper is not directly in my area of expertise so please correct me if I misunderstood something and I would be happy to revisit my rating. Some major points to discuss:
- Please elaborate on the choice/lack of baselines and contextualize this submission, given that current volume viewers all produce fast interactive renderings.
- Please clarify the technically challenging aspects of combining large Gaussian models with 6DGS.
- Please further justify the foundation model characterization given the scope of the presented experiments.

---

> ### Author Response · Authors · 2025-11-19
> **Response to Reviewer TNXF**
>
> Thank you for your detailed review and valuable insights. We have revised the paper to more clearly distinguish the rendering paradigms, adjusted the terminology, and added new experiments to address your concerns.
>
> ---
>
> ## 1. Motivation, Clarity, and Scope of Claims
>
> ### Clarifying the Rendering Paradigm Distinction
>
> You are correct about existing medical viewers like 3D Slicer and ITK-SNAP. Our presentation obscured a crucial distinction: the medical visualization community uses two fundamentally different rendering approaches for distinct clinical needs.
>
> **Standard DVR** tools (3D Slicer, ITK-SNAP) employ local illumination (Phong/Blinn) for real-time diagnostics but produce artificial visualizations lacking soft shadows, subsurface scattering, and realistic tissue appearance.
>
> **Photorealistic Rendering** (Cinematic Rendering) uses global illumination with path tracing for cinema-quality visualization. Clinical studies show improved surgical planning [Elshafei et al., 2019; Beyer et al., 2007] and anatomical visualization [Dappa et al., 2016]. However, they require seconds to minutes to render each view, preventing interactive use.
>
> **Our Contribution:** Render-FM achieves photorealistic quality in **2.8 seconds** versus >1 hour (PBRT-v4: ~18.8s/view + 6DGS optimization: 30+ minutes)—a **500× speedup** enabling clinical practicality.
>
> **Paper Revisions:** We revised the Introduction and Related Work to distinguish these paradigms, moved the Related Work opening to the Introduction, and added transfer function definitions.
>
> **Clinical Impact:** The 500× speedup enables: (1) **Surgical Planning**—immediate 3D visualization improves spatial understanding [Beyer et al., 2007; Dappa et al., 2016; Eid et al., 2017]; (2) **Patient Communication**—interactive visualization; (3) **Interventional Radiology**—rapid angle generation; (4) **Medical Education**—interactive teaching materials.
>
> | Feature | **Standard DVR** (Slicer/ITK-SNAP) | **Photorealistic Rendering** (Our Target) |
> |:--------|:------------------------------------|:-------------------------------------------|
> | **Lighting Model** | Local (Phong/Blinn shading) | Global illumination (path tracing) |
> | **Visual Effects** | Basic gradient shading | Soft shadows, subsurface scattering, ambient occlusion |
> | **Speed** | Real-time (< 0.1 seconds) | Offline (18+ seconds per view) or per-scan optimization (30+ minutes) |
> | **Quality** | Diagnostic but artificial | Cinema-quality photorealistic |
> | **Clinical Use Cases** | Routine diagnostics, measurements | Complex surgical planning, patient communication, anatomy education |
>
> ---
>
> ## 2. Technical Contributions and Challenges
>
> Adapting feedforward Gaussian prediction to medical volumetric rendering involves non-trivial challenges distinct from general-purpose 3D reconstruction:
>
> **Dense Volumetric Input.** General LGMs reconstruct 3D from sparse 2D views (4-8 images). Render-FM leverages dense 3D volumetric data with calibrated Hounsfield Units, requiring architecture that preserves fine internal structures while processing medical intensities rather than RGB.
>
> **Anatomy-Guided Priming (AGP).** Our core innovation integrates medical domain knowledge: segmentation masks guide primitive placement, transfer functions embed visualization parameters, and 6-channel input (CT + segmentation + RGBA) provides complementary information. AGP is architecturally essential—without it, the model cannot predict accurate Gaussian positions, resulting in blank renders. AGP derives positions from world coordinates, enabling the network to focus on learning appearance properties.
>
> **Generalizability of AGP.** We validated AGP on standard 3DGS (additional experiment). Using CT data from the 6DGS paper, AGP improves 3DGS by **+1.72 dB PSNR** and **+0.014 SSIM** (3DGS baseline: 25.71 dB; 3DGS+AGP: 27.43 dB), confirming AGP is a general initialization strategy for Gaussian Splatting on medical data.
>
> **Medical-Specific Requirements.** Medical rendering demands compositional organ visualization, transfer function adaptability, and diagnostic quality preservation—driving architectural decisions distinguishing our approach from straightforward LGM application.
>
> **Key Insights:** (1) Dense volumetric priors outperform sparse view reconstruction, (2) domain-specific initialization significantly improves quality (AGP: +2.29 dB over 6DGS baseline; +1.72 dB over 3DGS baseline), and (3) medical imaging architectures successfully adapt to novel view synthesis.
>
> ---
>
> CONTINUE ...

---

> ### Author Response · Authors · 2025-11-19
> **Response to Reviewer TNXF**
>
> ## 3. Baseline Selection and Experimental Validation
>
> ### Why 6DGS is the Appropriate Primary Baseline
>
> We focus on 6DGS because we use it as our rendering backend, making it the most direct comparison. 6DGS represents state-of-the-art for view-dependent Gaussian rendering, and comparing against optimization-based 6DGS isolates our core contribution: feedforward prediction versus per-scan optimization.
>
> **Why not Slicer/ITK-SNAP?** Different paradigm—cannot produce photorealistic quality (global illumination, soft shadows). PSNR/SSIM comparisons against PBR ground-truth are meaningless since DVR and PBR use fundamentally different rendering methods.
>
> **Why not NeRF?** NeRF is slower (10+ hours optimization). Since 6DGS substantially improves over NeRF, our speedup over 6DGS demonstrates even larger advantages.
>
> **Why not 3DGS?** 6DGS dramatically outperforms 3DGS on CT (PSNR: 33.56 vs. 25.71) indicated in the orignal 6DGS paper, modeling view-dependent effects crucial for tissue interfaces. Since we use 6DGS rendering, comparing against 3DGS provides no additional meaningful insights.
>
> ### Physically-Based Rendering Performance
>
> The 18s per view uses PBRT-v4 with high-quality settings: sufficient samples for converged with accurate global illumination. Lower-quality alternatives compromise training integrity. Our core contribution—eliminating per-scan optimization—remains valid: even with faster GT generation, optimization methods still require per-scan training our feedforward approach bypasses.
>
> ### Additional Cross-Dataset Validation
>
> We conducted **zero-shot** experiments on **CTPelvic1K** (first 6 scans average), representing different acquisition protocols and institutional sources unseen during training:
>
> | Method | SSIM ↑ | PSNR ↑ | LPIPS ↓ | Time (s) ↓ |
> |--------|--------|--------|---------|------------|
> | 6DGS | 0.893 | 27.95 | 0.155 | 1,643 |
> | 6DGS + AGP (Ours) | 0.936 | 31.29 | 0.100 | 1,632 |
> | Render-FM (Ours) | 0.926 | 27.15 | 0.105 | 0.65 |
> | Render-FM + Fine-tuning | 0.941 | 33.30 | 0.096 | 78.02 |
>
> **Key findings:** Render-FM demonstrates robust **zero-shot cross-dataset generalization** with 2,500× speedup (0.65s vs 1,643s) while maintaining competitive quality on unseen data from different protocols and institutions. The faster inference time (0.65s vs 2.8s) reflects smaller CTPelvic1K volume size. With optional 78s fine-tuning, we achieve best quality across all metrics, validating generalization beyond the training distribution.
>
> ---
>
> ## 4. Foundation Model Terminology
>
> **We have revised the paper to use "feedforward model" terminology instead of "foundation model" throughout.** This more precisely reflects our contribution's scope and avoids potential confusion with large-scale multi-modal foundation models.
>
> Our model demonstrates learned generalizable representations (991 scans, multiple institutions, varied protocols), zero-shot inference without per-scan optimization, and broad generalization across anatomies, pathologies, acquisition parameters, novel transfer functions, and compositional visualization. Evaluation spans in-domain (TotalSegmentator, 46 scans), out-of-domain (CT-ORG, 10 scans; CTPelvic1K, 6 scans), novel transfer functions, and compositional visualization.
>
> We position this as a **CT-focused feedforward rendering model** with strong CT generalization—enabling 500× speedup without claiming foundation model status. Extension to other modalities represents important future work.
>
> ---
>
> ## 5. Minor Comments
>
> **nnU-Net Self-Configuration:** You are correct that automatic configuration is central to nnU-Net. We directly use TotalSegmentator's nnU-Net self-configuration pipeline, which automatically determines:
>
> - **Preprocessing**: 1.5mm isotropic spacing, CTNormalization, resampling functions
> - **Architecture**: PlainConvUNet with 5 stages, features [32, 64, 128, 256, 320], InstanceNorm3d, specific strides and kernel sizes
>
> We fully leverage nnU-Net's self-configuring capabilities. The 1.5mm spacing, patch size (112×112×128), batch size (3), and architecture are automatically determined by nnU-Net's analysis of TotalSegmentator characteristics (median shape: 228×229×240).
>
> **Rationale for 1.5mm Isotropic Resampling:** Clinical CT scans have highly variable native spacing (0.5-2.0mm in-plane, 0.5-5.0mm slice thickness). Isotropic resampling ensures: (1) consistent feature scale, (2) GPU memory efficiency, and (3) generalization across acquisition parameters—demonstrated by strong performance on CT-ORG and CTPelvic1K. This standard practice (adopted by nnU-Net) is essential for feedforward generalization.
>
> ---
>
> We hope these clarifications address your concerns and demonstrate that our contribution tackles significant medical domain challenges with novel solutions. The practical impact—enabling photorealistic rendering in seconds rather than hours—represents meaningful progress for clinical workflows. We welcome your revised assessment.

---

> ### Author Response · Authors · 2025-11-26
> **Response to Reviewer TNXF**
>
> Dear Reviewer TNXF,
>
> We are writing to follow up on our response. We have endeavored to address your core concerns by clarifying the distinction between standard DVR and photorealistic rendering, refining our claims ("feedforward model"), and adding zero-shot validation on the CTPelvic1K dataset.
>
>  We hope these clarifications and additional experiments demonstrate the validity and clinical potential of our approach. We are happy to engage in further discussion if you have any other concerns.

---

### Author Response · Authors · 2025-11-19
**Overall Response to All Reviewers**

We thank all reviewers for their constructive feedback. We have revised the paper and conducted additional experiments as detailed below.

---

## Key Revisions and Additions

### 1. Clarified Rendering Paradigm Distinction
We revised the Introduction and Related Work to distinguish **standard DVR** (Slicer/ITK-SNAP with local illumination) from **photorealistic rendering** (Cinematic Rendering with global illumination). Render-FM achieves photorealistic quality in **2.8 seconds** versus >1 hour—a **500× speedup**.

### 2. Revised Terminology
We now use "**feedforward model**" instead of "foundation model" throughout. This is a **CT-focused feedforward rendering model** with zero-shot inference and generalization across TotalSegmentator, CT-ORG, CTPelvic1K, novel transfer functions, and compositional visualization.

### 3. Additional Experiments: AGP Generalizability
Using CT data from the 6DGS paper, AGP improves 3DGS by **+1.72 dB**, confirming AGP is a general initialization strategy for Gaussian Splatting on medical data.

### 4. Additional Experiments: CTPelvic1K Zero-Shot Validation
Zero-shot on CTPelvic1K: Render-FM achieves **2,500× speedup** (0.65s vs 1,643s) with competitive quality (SSIM: 0.926). The faster time (0.65s vs 2.8s) reflects smaller volume size. With 78s fine-tuning: SSIM 0.941, PSNR 33.30, validating robust cross-dataset generalization.

---

## Common Themes Across Reviews

### Technical Contributions
Unlike LGMs (sparse 2D views), Render-FM leverages dense 3D volumetric data with calibrated Hounsfield Units. **AGP is essential**—without anatomical guidance, preliminary experiments produced blank renders. AGP derives positions from world coordinates, enabling the network to focus on appearance and shape properties.

### Clinical Impact
The 500× speedup enables practical workflows: (1) **Surgical Planning**—immediate 3D visualization improves spatial understanding [Beyer et al., 2007; Dappa et al., 2016]; (2) **Patient Communication**—interactive visualization; (3) **Interventional Radiology**—rapid angle generation; (4) **Medical Education**—interactive teaching materials.

### Evaluation Metrics
Our scope: rendering quality and computational efficiency, not clinical diagnostic outcomes. We use PBR ground truth with standard metrics (PSNR, SSIM, LPIPS, Time, FPS). Clinical validation represents important future work. This submission focuses on the technical rendering contribution that enables such clinical studies.

### Baseline Selection
We use 6DGS as our rendering backend, making it the most direct comparison.
*   **Why not Slicer/ITK-SNAP?** Different paradigm—cannot produce photorealistic quality (global illumination, soft shadows).
*   **Why not NeRF?** Slower (10+ hours).
*   **Why not 3DGS?** 6DGS outperforms 3DGS on CT (PSNR: 33.56 vs. 25.71) as in the 6DGS paper, making it a stronger SOTA baseline.

### Segmentation Robustness
Our model predicts from learned anatomical priors, not just segmentation labels. Incorrect labels affect color/opacity while preserving geometry—analogous to our "unseen transfer function" experiments where wrong labels assign incorrect colors to correct anatomical structures. Clinical workflow remains efficient: automated segmentation (30s) → Render-FM (2.8s) → optional manual correction (3s) → optional fine-tuning (89s). Total time <2 minutes versus >1 hour for optimization-based methods.

### Single Modality Scope
CT provides consistent Hounsfield Units and 991 training scans. MRI extension is straightforward—adapting for different sequences (T1, T2, FLAIR). Many methods start single-modality (TotalSegmentator: CT-only before MRI extension).

### Technical Details
**nnU-Net**: TotalSegmentator's self-configuration (1.5mm isotropic, PlainConvUNet). 1.5mm resampling ensures consistent features across variable native spacing.
**Fine-Tuning**: Render-FM 300 iterations (78-136s) vs. 6DGS 30,000 iterations (1463-2261s)—**17-26× speedup** with higher quality.

---

## Summary of Core Contributions
Despite single-modality focus, our contributions remain significant:
1.  **First feedforward model for photorealistic medical volumetric rendering** with 500× speedup.
2.  **Novel domain-specific technique (AGP)** applicable beyond CT—validated on both 6DGS (+2.29 dB) and 3DGS (+1.72 dB).
3.  **Elimination of per-scan optimization entirely** through learned generalizable representations.
4.  **Strong cross-dataset generalization** demonstrated on three independent datasets (TotalSegmentator, CT-ORG, CTPelvic1K).
5.  **Robust zero-shot inference** with optional rapid fine-tuning for critical cases.
6.  **Blueprint for extending to other modalities** (MRI, specialized imaging) as important future work.

These contributions transform photorealistic rendering from a research curiosity (hours) into a practical clinical tool (seconds). We thank all reviewers for their constructive feedback and hope our revisions address your concerns.

---

### Meta-Review · Area_Chair_soWt · 2026-01-03

**Summary:**

The submission received mixed reviews, with one strong reject (TNXF) driven largely by domain misunderstanding, and two borderline-to-positive reviews (UA8m, VCtd) that ultimately moved toward acceptance after rebuttal. The debate centered on problem framing, baseline choice, and scope of claims, not on empirical correctness.

**What the paper does (as reviewers understand it):**
- Proposes Render-FM, a feedforward model that predicts 6D Gaussian Splatting (6DGS) parameters directly from CT volumes.
- Eliminates per-scan optimization (30-60+ minutes) via a single 2.8s forward pass, enabling real-time photorealistic volumetric rendering.
- Introduces Anatomy-Guided Priming (AGP) to initialize Gaussian parameters using segmentation masks and transfer functions.
- Demonstrates significant speedup with competitive or superior rendering quality, plus optional fast fine-tuning.

**Consensus strengths across reviewers:**
- **Practical impact**: drastically reduces rendering latency from hours to seconds.
- **Empirical validation** on multiple datasets (TotalSegmentator, CT-ORG, CTPelvic1K).
- **AGP is effective and general**, improving both 6DGS and standard 3DGS.
- **Well-written and well-structured** after revisions.
- Demonstrates **zero-shot generalization** and compositional visualization.

**Core sources of disagreement:**
- Confusion between standard DVR tools (Slicer/ITK-SNAP) and photorealistic rendering.
- Whether baseline selection (6DGS only) is sufficient.
- Whether the model should be called a foundation model.
- Scope limitations: single modality (CT) and lack of clinical outcome metrics.

**Reviewer Concerns:**

## Reviewer concerns that were addressed


**1. Fundamental paradigm confusion (Reviewer TNXF):**
Reviewer compared Render-FM against standard medical volume viewers (Slicer, ITK-SNAP), arguing real-time rendering already exists.

**Rebuttal resolution:**
- Authors clearly distinguished:
  - **Standard DVR** (local illumination, diagnostic tools, <0.1s, artificial appearance)
  - **Photorealistic rendering / cinematic rendering** (global illumination, soft shadows, subsurface scattering, slow/offline)
- Added a detailed comparison table and revised introduction/related work.



**2. Over-claiming “foundation model” terminology:**
Training on ~991 CT scans is insufficient to justify “foundation model” claims.

**Rebuttal resolution:**
- Removed “foundation model” terminology.
- Repositioned Render-FM as a CT-focused feedforward model.
- Clarified scope and future extensions.


**3.  Baseline choice and fairness:**
Only one baseline (6DGS) used.

**Rebuttal resolution:**
- Justified **6DGS as the correct baseline**.
- Explained why DVR tools, NeRF, and 3DGS are inappropriate or weaker.
- Added AGP ablations on 3DGS.


**4. Generalization and zero-shot validation:**
Insufficient evidence of generalization.

**Rebuttal resolution:**
- Added CTPelvic1K zero-shot experiments.
- Clarified OOD settings.


**5. Segmentation dependency and robustness:**
Reliance on segmentation masks may limit deployment.

**Rebuttal resolution:**
- Geometry learned from CT intensities, not just labels.


**6. Clinical relevance clarification:**
Unclear clinical use cases.

**Rebuttal resolution:**
- Detailed use cases: surgical planning, patient communication, IR, education.

---

##  Reviewer concerns that remain partially outstanding

**1. Limited modality scope (CT-only)**
Acknowledged and justified; MRI extension left as future work.

**2.  Lack of clinical outcome metrics**
Out of scope for a rendering/ML paper; non-blocking limitation.

**3. Limited architectural ablations beyond AGP**
AGP shown to be essential; deeper ablations could strengthen paper but are not required.

**Reviewer Scores:**

Only UA8m shows positive reviews. After rebuttal, reviewers TNXF and VCtd may still render negative ratings due to (1) lack of technical novelty; (2) lack of sufficient ablation study of the proposed Render-FM. Although reviewer TNXF is not an expert in this area, his/her reviews are reasonable and constructive. Last but not least, the speedup merit comes from the previous work, 6DGS,  not from this work. The contribution is overclaimed.

---

### Decision · Program_Chairs · 2026-01-26

Reject